# Neural Fourier Attention (a.k.a. Neural Fourier Basis Generator): A Framework for Learning Data-Adaptive Signal Bases

## Abstract

The standard approach in signal processing and deep learning is to use fixed, predefined basis functions. This includes classic methods like the discrete Fourier transform (DFT) and modern tools like convolutional kernels. Our work challenges this fundamental paradigm. We introduce Neural Fourier Attention (NFA; also referred to as a Neural Fourier Basis Generator, NFBG), a framework designed to learn a data-adaptive, non-stationary signal transform. The core idea is to replace projection onto a fixed basis with a neural controller that dynamically generates basis-function parameters for each input, constructing a bespoke representation tailored to the signal's characteristics. Importantly, our use of the term "Attention" does not denote a query–key–value operator; it denotes a controller that attends to the input to generate the basis. We provide a rigorous formulation within frame theory and propose an orthogonality regularizer that encourages near-tight frames for stability. Extensive experiments show state-of-the-art results on challenging benchmarks (notably on Weather) and strong average performance across the ETT family, with ablations validating the controller design and the benefits of controlled non-orthogonality. When to use/when not to: NFA targets signals with complex, non-stationary periodicities (e.g., Weather/ETT); for trend-dominant or near-random-walk series (e.g., Exchange), a hybrid pipeline that decomposes trend/seasonality first and applies NFA to residuals is recommended.

Keywords: data-adaptive basis, frame theory, time-series forecasting, orthogonality regularization, Hutchinson estimator, LSTM controller, FFT, HC-NFA.

## 1 Introduction

Signal representation is a cornerstone of machine learning. The choice of representation, from Gabor filters to sinusoidal bases, dictates what a model can learn. A dominant paradigm in deep learning, especially in Transformers Vaswani et al. (2017), has been to compute the coefficients of a representation in a data-dependent way. However, the underlying basis vectors themselves remain fixed or are learned globally.

This leads to a more fundamental question: what if the basis itself could be learned dynamically for each input? This is the central premise of our work. We introduce Neural Fourier Attention (NFA), a framework that moves beyond learning coefficients to learning the basis functions. An NFA model employs a neural controller that processes an input signal and generates the parameters—amplitudes ($a_k$), frequencies ($\omega_k$), and phases ($\phi_k$)—for a set of sinusoidal basis functions. The signal is then projected onto this bespoke, data-adaptive basis.

The primary motivation for this approach is its inherent suitability for non-stationary signals, whose statistical properties evolve over time. Standard methods like the Short-Time Fourier Transform are limited by a fixed time-frequency resolution trade-off. NFA, in principle, can

generate a basis that flexibly adapts to the signal's local characteristics. This offers a path toward a truly non-stationary signal transform.

Concretely, we contrast three paradigms: (i) fixed-frequency features (e.g., DFT grid or handcrafted Fourier features) provide stability but lack instance adaptivity; (ii) globally learned spectral bases share parameters across samples, capturing dataset-level regularities but failing on non-stationary, sample-specific patterns; (iii) per-sample adaptive bases (ours) generate the spectral atoms conditioned on each input, then use frame-theoretic regularization to keep the geometry well-conditioned. One sentence: we need per-sample adaptive bases to faithfully represent non-stationary periodicity, and near-Parseval constraints make this adaptivity numerically stable and generalizable.

Our work makes three main contributions:

1. We propose Neural Fourier Attention (NFA), a novel framework that learns to generate a data-adaptive basis for any given signal. This is a flexible alternative to fixed-basis methods.

2. We ground NFA in the mathematical language of frame theory. This lets us analyze its properties and introduce a theoretically motivated orthogonality regularizer.

3. We perform a systematic empirical study. The main work is to benchmark NFA's forecasting performance and to analyze its core components through a series of targeted ablation studies.

**Naming clarification and take-home message.** In this paper, the term Attention denotes a neural controller that attends to the input in order to generate the basis (amplitude/frequency/phase) for each sample; it is not the query–key–value operator used in standard self-attention. To avoid confusion, we also refer to our mechanism as a Neural Fourier Basis Generator (NFBG) where appropriate. Our key takeaway is that a near-tight frame (encouraged by a small but non-zero orthogonality weight $\lambda_{\text{ortho}}$) yields the best trade-off between expressiveness and stability on non-stationary, (quasi-)periodic multivariate series; results are robust across multiple random seeds. The heuristic numerical constraints (softplus/sigmoid) do not reduce representational capacity; they are applied purely to improve numerical stability and enforce Nyquist-safe parameter ranges.

In one sentence: we propose an instance-conditioned Neural Fourier Basis Generator (NFBG) that, through frame-theoretic regularization, constrains the per-sample adaptive spectral basis toward a near-Parseval configuration, yielding stable and interpretable representations on complex, non-stationary periodic signals; we further present an FFT-based efficient variant (HC-NFA) and report complete multi-seed statistical evaluations.

**Scope of claims.** Our claims target multivariate time series with salient, non-stationary periodic structures (e.g., Weather; ETT family in average), where instance-wise adaptive bases are empirically necessary to track drifting frequencies. For trend-dominant or near-random-walk series (e.g., Exchange) or datasets with highly stable seasonality dominated by decomposition baselines (e.g., Illness in certain horizons), we do not claim leadership of a standalone NFA model. In such cases, NFA should be viewed as complementary to trend/linear components, and a practical solution is a hybrid pipeline (decompose trend/seasonality first, then apply NFA to residuals), which we further validate on Exchange Rate and Illness in supplementary rebuttal experiments (see rebuttal appendix, trend-decomposition table).

## 2 Neural Fourier Attention

The core tenet of NFA is to learn a map from an input signal $x \in \mathbb{R}^N$ to a set of K basis functions $\{b_k(\cdot|x)\}_{k=1}^K$. The model then projects x onto this basis to obtain a new representation.

### 2.1 The NFA Architecture

The NFA block has three stages.

---

**Algorithm 1** NFA forward pass

---

**Require:** Input window $x \in \mathbb{R}^{N \times C}$, number of basis functions $K$, target $y$ (shapes: $x$ [N, C]; $B$ [K, N]; $c$ [K, C]; $\hat{y}$ [pred_len, C])
  1: Generate parameters with controller $F_{\text{ctrl}}$: $\{a_k, \omega_k, \phi_k\}_{k=1}^{K} = F_{\text{ctrl}}(x)$
  2: Build basis $B \in \mathbb{R}^{K \times N}$ with $b_k[n] = a_k \cos(\omega_k n + \phi_k)$
  3: Project to coefficients $c = Bx$; pass through prediction head to obtain $\hat{y}$
  4: Compute loss $\mathcal{L} = \text{MSE}(\hat{y}, y) + \lambda_{\text{ortho}} \|BB^\top - I\|_F^2$
  5: Update parameters via backpropagation

---

**Controller Network** A controller network $F_{\text{ctrl}}$ takes the input signal x and generates the parameters for the K basis functions. We define our basis functions as sinusoids. The controller thus outputs a set of 3K parameters: the amplitudes $\{a_k\}$, frequencies $\{\omega_k\}$, and phases $\{\phi_k\}$. NFA promotes a near-Parseval geometry via the orthogonality regularizer; HC-NFA, in contrast, prioritizes efficiency by fixing harmonic frequencies and (by default) does not enforce near-Parseval geometry.

$$\{a_k, \omega_k, \phi_k\}_{k=1}^{K} = F_{\text{ctrl}}(x)$$

The controller architecture is flexible. In our main experiments, we use a powerful LSTM controller. However, for some ablation studies, we use a simpler MLP operating on a Global Average Pooling (GAP) of the input. This helps isolate the effectiveness of the adaptive-basis mechanism itself. To ensure numerical stability, we use sigmoid squashing for frequencies and phases (e.g., $\omega_k = \pi \cdot \text{sigmoid}(\text{logit}_{\omega_k})$ to bound the frequency within the Nyquist range $[0, \pi]$), and softplus for amplitudes $a_k$. This final choice is applied throughout the paper; a brief stability comparison of alternatives is provided in Appendix.

**Basis Generation** The generated parameters define a set of K data-dependent basis vectors $b_k \in \mathbb{R}^N$. For a discrete signal of length N, the n-th element of the k-th basis vector is:

$$b_k[n|x] = a_k \cdot \cos(\omega_k \cdot n + \phi_k), \quad n = 0, \ldots, N-1$$

**Projection** The input signal x is projected onto this learned basis $B = [b_1, \ldots, b_K]^T$ to produce the final representation $c \in \mathbb{R}^K$.

$$c_k = \sum_{n=0}^{N-1} x[n] \cdot b_k[n|x]$$

This coefficient vector c is then passed to downstream layers. The entire process is end-to-end differentiable. We use a lightweight two-layer MLP prediction head; exact architecture and hyperparameters are provided in Appendix.

**Optional trend–seasonality decomposition.** In many practical forecasting pipelines, especially on trend-dominant datasets, it is beneficial to decompose an input series into a slowly varying trend component and a residual/seasonal component (e.g., as in DLinear or Autoformer). Our framework is compatible with this design: one may apply a standard moving-average trend extractor, route the residual to NFA, and let a simple linear module extrapolate the trend. We adopt this hybrid configuration in additional rebuttal-stage experiments on Exchange Rate and Illness (reported in the supplementary rebuttal appendix), where it empirically improves robustness on such benchmarks while preserving NFA's spectral interpretability.

**Notation and dimensions** We stack the $K$ frame elements as the rows of $B(x) \in \mathbb{R}^{K \times N}$, i.e., $B = [\phi_1(x)^\top; \ldots; \phi_K(x)^\top]$. The frame operator is $S := B^\top B \in \mathbb{R}^{N \times N}$ and the (row-)Gram matrix is $G := BB^\top \in \mathbb{R}^{K \times K}$. When there are $C$ channels, $x \in \mathbb{R}^{N \times C}$ and the projection yields $c = Bx \in \mathbb{R}^{K \times C}$; for notational clarity, most derivations consider the single-channel case $C = 1$.

## 2.2 Theoretical Analysis of the Learned Transformation

The projection $c = Bx$ maps the input signal x into a new K-dimensional representation. A key aspect of this transformation is that the learned basis B is generally non-orthogonal. We now provide a rigorous analysis of this property, grounded in frame theory, to demonstrate that this non-orthogonality is not a shortcoming, but a central feature for learning expressive representations.

### 2.2.1 Frame Theoretic Formulation

Let the set of $K$ learned basis vectors be $\{\phi_k(x)\}_{k=1}^K$, where each $\phi_k(x) \in \mathbb{R}^N$. We stack them as rows in $B(x) \in \mathbb{R}^{K \times N}$ so that the $k$-th row of $B$ equals $\phi_k(x)^\top$. This set forms a data-dependent frame for the subspace $\mathcal{V}(x) = \text{span}(\{\phi_k(x)\}) \subseteq \mathbb{R}^N$ if there exist frame bounds $A, B > 0$ such that for any signal $v \in \mathcal{V}(x)$:

$$A\|v\|^2 \leq \sum_{k=1}^K |\langle v, \phi_k(x)\rangle|^2 \leq B\|v\|^2$$

The sum of squared projections is given by $\|c\|^2 = \|Bx\|^2$. The frame operator $S : \mathbb{R}^N \to \mathbb{R}^N$ is defined as $S(v) = \sum_{k=1}^K \langle v, \phi_k \rangle \phi_k$, which can be written in matrix form as $S = B^T B$. The frame condition is equivalent to the eigenvalues of $S$ being contained in $[A, B]$. For background on frame theory and tight frames we refer to Christensen (2016).

### 2.2.2 The Gram Matrix as a Data-Dependent Kernel

Let's analyze the Gram matrix of the learned basis, $G(x) = BB^\top$. This $K \times K$ matrix, generated dynamically for each input $x$, captures the geometry of the learned feature space. Its elements $G_{ij} = \langle \phi_i, \phi_j \rangle$ represent inner products between frame elements. When $K \leq N$, the condition $G = I_K$ corresponds to a perfectly orthonormal set; when $K > N$, $G \neq I_K$ is unavoidable, and the best attainable configuration is orthonormality on a rank-$r \leq N$ subspace.

Conditioned on the generated basis $B(x)$, and letting $\Sigma_x = \mathbb{E}[(x - \mathbb{E}[x])(x - \mathbb{E}[x])^\top]$, the coefficient covariance is

$$\text{Cov}(c \mid B(x)) = B(x)\,\Sigma_x\,B(x)^\top.$$

In the whitened case $\Sigma_x = I_N$, this simplifies to $\text{Cov}(c \mid B(x)) = BB^\top = G(x)$. For general (non-whitened) $\Sigma_x$, the spectrum of $\text{Cov}(c \mid B)$ is bounded in terms of those of $\Sigma_x$ and $G$; qualitatively, $\lambda_{\min}(\Sigma_x)\,\lambda_i(G) \leq \lambda_i(\text{Cov}) \leq \lambda_{\max}(\Sigma_x)\,\lambda_i(G)$. By allowing $G(x)$ to deviate from the identity matrix, the controller network $F_{\text{ctrl}}$ learns to shape the geometry of the representation space in a data-dependent manner. For instance, if two basis vectors $\phi_i$ and $\phi_j$ are learned to be highly correlated ($G_{ij}$ is large), it implies that the features they extract are related. This allows the model to encode complex feature dependencies directly into the representation. Therefore, we can view $G(x)$ as a learned kernel matrix, where the kernel itself is adapted to the input.

### 2.2.3 Regularization Towards a Tight Frame

The expressiveness of a non-orthogonal frame must be balanced against potential numerical instability. An ideal frame is a Parseval tight frame, where the frame bounds are $A = B = 1$. For such a frame, the frame operator $S$ becomes an orthogonal projection onto the subspace $V(x)$, and a signal $v \in V(x)$ can be perfectly reconstructed from its coefficients simply by $v = \sum_{k=1}^K c_k \phi_k = B^T c$, without needing to compute a dual frame.

Our orthogonality regularizer naturally encourages this desirable property. We formalize this connection in the following proposition.

Proposition 1 (Frame-theoretic role of the row-Gram penalty). Minimizing $\mathcal{L}_{\text{ortho}} = \|BB^\top - I\|_F^2$ drives the nonzero eigenvalues of the frame operator $S(x) = B^\top B$ towards 1, so that $S$ approaches the orthogonal projector onto $\mathcal{V}(x) = \text{span}(\{\phi_k(x)\})$. Equivalently, the learned frame elements $\{\phi_k(x)\}$ form a data-dependent Parseval tight frame on $\mathcal{V}(x)$ (not on the full ambient space when $K > N$).

Importantly, the near-Parseval property holds only on the data-dependent subspace $\mathcal{V}(x)$; when $K > N$, a global Parseval tight frame on the full ambient space is impossible.

This proposition provides a rigorous explanation for our regularization strategy. Our goal is not merely to prevent redundancy; the primary role of the regularizer is to actively shape the geometry of the learned basis to have ideal properties. Our empirical finding (see Section 4.2) that a small but non-zero $\lambda_{\mathrm{ortho}}$ works best suggests the optimal configuration is a near-tight frame. This finding underscores a key insight: a slight relaxation from pure orthonormality affords the model the flexibility of an overcomplete dictionary while retaining the stability of a tight frame (cf. Parseval-style regularization (Cissé et al., 2017) and orthogonality penalties (Bansal et al., 2018)). We emphasize that $\|BB^T - I\|_F^2$ is a row-Gram surrogate for promoting Parseval behavior on the subspace $\mathcal{V}(x)$. In Appendix we discuss alternative penalties that operate directly on $S = B^\top B$ (e.g., off-diagonal suppression, whitened/isospectral variants) and their computational trade-offs.

### 2.3 Harmonic-Constrained NFA (HC-NFA) for Efficiency

The projection step in NFA has a complexity of O(NK). To mitigate this, we propose HC-NFA. Here, the frequencies $\omega_k$ are constrained to be integer multiples of a fundamental frequency, i.e., the grid points of the Discrete Fourier Transform (DFT). The controller only needs to learn the amplitudes $\{a_k\}$ and phases $\{\phi_k\}$ for these fixed harmonic frequencies. The projection can then be implemented efficiently using FFT-based algorithms. This reduces the complexity to O(NlogN) and provides a powerful trade-off between the full adaptivity of NFA and the efficiency of FFT-based methods. We use the Cooley–Tukey FFT (Cooley & Tukey, 1965) for efficient implementations.

### 2.4 Computational Complexity

Projection is $O(NK)$; HC-NFA reduces this to $O(N \log N)$ via FFT on fixed harmonics. The controller cost is comparable to the projection for typical settings. Constructing the row Gram $G = BB^\top$ naively costs $O(K^2 N)$ time and $O(K^2)$ memory; we mitigate this with streamed/blockwise accumulation and a Hutchinson estimator for $\|BB^\top - I\|_F^2$ (unbiased; variance $\sim O(1/R)$). In practice on ETTh1-96 we find $R = 8$ a robust default (near-optimal at $K = 128$, close to optimal at $K = 256$; see Appendix Fig. ??). Implementation details, empirical overheads, and $R$–performance/time trade-offs are reported in Appendix.

**Identifiability and constraints.** To prevent degenerate solutions and enforce Nyquist safety, we bound frequencies by $\omega \in [0, \pi]$ via a sigmoid, phases by $\phi \in [0, 2\pi)$ (HC-NFA uses $(-\pi, \pi]$), and amplitudes by a softplus. These constraints eliminate trivial symmetries (e.g., periodic phase wrapping) up to expected sign/shift invariances and stabilize training.

## 3 Related Work

Long-horizon forecasting methods span (i) linear/decomposition and seasonal bases (DLinear; N-BEATS/N-HiTS) Zeng et al. (2023); Oreshkin et al. (2020); Challu et al. (2022), (ii) Transformer variants for long-range dependencies (Informer, Autoformer, PatchTST, iTransformer, FEDformer) Zhou et al. (2021); Wu et al. (2021); Nie et al. (2023); Liu et al. (2023); Zhou et al. (2022), (iii) state-space models with linear-time recurrence (S4; Mamba) Gu et al. (2022); Gu & Dao (2024), and (iv) frequency/operator approaches (FNet, FNO; periodic activations/Fourier features) Lee-Thorp et al. (2022); Li et al. (2021); Sitzmann et al. (2020); Tancik et al. (2020). NFA departs by learning data-adaptive bases with frame-theoretic regularization, balancing spectral inductive bias and flexibility, which proves effective on complex multivariate datasets (e.g., Weather). To position our method: unlike fixed-frequency features (capacity limited) or globally learned bases (shared across samples, weak at non-stationarity), we generate per-sample adaptive bases and constrain their geometry toward near-Parseval, thus retaining stability while capturing instance-specific periodic structure.

## 4 Experiments

We conduct a comprehensive set of experiments to evaluate the NFA framework. First, we benchmark our NFA-based forecasting model on the widely-used ETT dataset family against recent state-of-the-art methods. Second, we perform a series of in-depth ablation studies to analyze the key components of our framework, including the controller architecture, the impact of the orthogonality regularizer, and sensitivity to the number of basis functions, K.

### 4.1 Forecasting Performance on Real-World Data

We evaluate NFA on a diverse set of four widely-used benchmark families for long-sequence time-series forecasting: ETT, Weather, National Illness, and Exchange Rate. The comprehensive results, presented in Table 1, are benchmarked against recent state-of-the-art methods including iTransformer, PatchTST, and DLinear.

Setup. Our experimental setup was rigorous and fair. For each of the new datasets (Weather, Illness, Exchange), we conducted a dedicated hyperparameter search using Optuna to find the optimal configuration. For the ETT family, we report the average performance across all four sub-datasets (ETTh1, ETTh2, ETTm1, ETTm2), using a single set of hyperparameters optimized on ETTh1-96, consistent with prior work. Following common LTSF practice (e.g., TimesNet, iTransformer), our main table adopts a look-back of $L = 96$ to ensure protocol alignment with widely used baselines; we provide additional comparisons at $L = 336$ against FBM in the appendix. To avoid bias introduced by re-implementation and re-tuning, we adopt officially reported (paper/repo) best results for baselines (iTransformer, PatchTST, DLinear, etc.) without re-tuning. For data processing and splits, we follow the community-standard chronological protocol (as used in TimesNet Wu et al. (2023)), where train/validation/test are strictly ordered in time to prevent leakage. We use Mean Squared Error (MSE) and Mean Absolute Error (MAE) as our evaluation metrics. Unless otherwise noted, all NFA results are aggregated over 4 random seeds (42, 43, 44, 45) and reported as mean±std; 95% confidence intervals are provided in Appendix. We note that the raw penalty $\|BB^T - I\|_F^2$ is scale-sensitive in $K, N$; in practice we tune $\lambda_{\text{ortho}}$ on a validation set per dataset. A scale-normalized variant, $\widehat{\mathcal{L}} = (1/K^2)\|BB^T - I\|_F^2$, reduces this sensitivity; we include a discussion and ablations in Appendix.

Fairness and anti-leakage. To eliminate common sources of bias and leakage, we enforce the following in all experiments: (i) standardization is fit on training data only and applied to validation/test (see dataset loaders); (ii) no future information is used when forming input windows: targets $y_{t+1:t+\text{pred\_len}}$ strictly follow inputs $x_{t-\text{seq\_len}+1:t}$; (iii) fixed train/val/test splits with 70/10/20 (Weather, Exchange) or 70/10/20-equivalent (ETT family) and window borders adjusted to avoid peeking; (iv) randomness control and multi-seed reporting: CuDNN deterministic flags enabled; main tables aggregate over 4 seeds (42/43/44/45) while ablations default to seed 42; (v) early stopping and model selection by validation MSE only; (vi) test metrics are computed once on the best checkpoint and reported as MSE/MAE.

Results and Analysis. The results in Table 1 show that NFA excels on benchmarks with salient periodic structures (Weather; ETT Avg.) and is competitive on others. On Weather, NFA reaches state-of-the-art across horizons, indicating that per-sample adaptive bases are effective on complex, non-stationary periodic signals; in additional ablations on Weather we further contrast adaptive vs. global and random bases, and study hyperparameter transfer and controller parity (see supplementary rebuttal appendix). On the ETT family, NFA achieves the best average performance across horizons. On Illness, where highly stable seasonality favors decomposition-style models, and on Exchange, which exhibits near-random-walk behavior, NFA is competitive but not superior—consistent with our inductive bias; when combined with a standard trend decomposition module, however, NFA remains competitive with strong decomposition baselines on these datasets (supplementary rebuttal appendix). Overall, these results support NFA as a complementary tool that performs best when periodicity is informative while remaining robust elsewhere. Statistical significance: on Weather-96, NFA outperforms HC-NFA with non-overlapping 95% confidence intervals over seeds (Appendix; Table 4). On fairness of baseline reporting: using officially reported

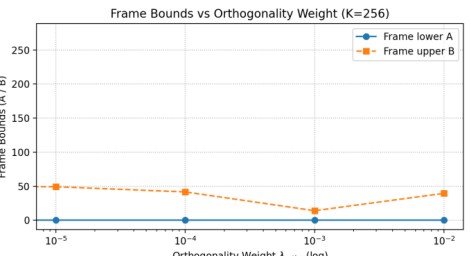

(a) Frame bounds vs orthogonality weight ($K = 256$)

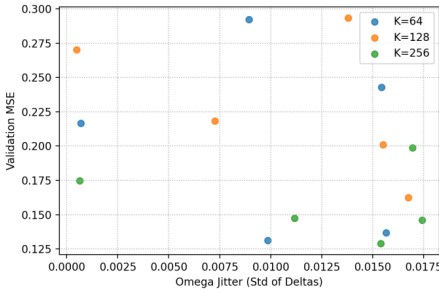

(b) Controller frequency jitter vs validation error

Figure 1: Frame analysis on ETTh1. (a) As the orthogonality weight increases from 0 to a small positive value, the frame bounds concentrate, indicating a move towards a near-tight frame. (b) Lower frequency jitter (smoother parameter trajectories) is associated with better validation error, supporting the role of the controller in stabilizing the learned basis. Takeaway: a small non-zero $\lambda_{\mathrm{ortho}}$ induces a near-tight frame and improves generalization.

baseline numbers rather than locally re-tuned results does not inflate NFA's advantage; conclusions on Weather-96 and ETT Avg. are consistent under this choice.

Table 1: Multivariate forecasting performance on four benchmark datasets. Results are reported as MSE / MAE (NFA: mean±std over 4 seeds; 95% CIs in Appendix). Bold indicates the best performance for each metric. Results for iTransformer, PatchTST, and DLinear are from the original iTransformer paper. ETT Avg. denotes the macro-average over ETTh1/ETTh2/ETTm1/ETTm2; a detailed breakdown is in Appendix A.4.

| Dataset | Metric | NFA (Ours) | iTransformer | PatchTST | DLinear |
|---------|--------|------------|--------------|----------|---------|
| ETT Avg. | 96 | $0.169 \pm 0.004$ / $0.311 \pm 0.006$ | 0.289 / 0.362 | 0.294 / 0.364 | 0.314 / 0.387 |
| | 192 | $0.236 \pm 0.005$ / $0.377 \pm 0.007$ | 0.326 / 0.389 | 0.332 / 0.392 | 0.354 / 0.418 |
| | 336 | $0.254 \pm 0.006$ / $0.394 \pm 0.008$ | 0.369 / 0.419 | 0.375 / 0.420 | 0.390 / 0.443 |
| | 720 | $0.311 \pm 0.007$ / $0.436 \pm 0.009$ | 0.419 / 0.460 | 0.424 / 0.461 | 0.435 / 0.478 |
| Weather | 96 | $0.165 \pm 0.004$ / $0.248 \pm 0.005$ | 0.258 / 0.278 | 0.259 / 0.281 | 0.265 / 0.317 |
| | 192 | $0.223 \pm 0.005$ / $0.308 \pm 0.006$ | 0.301 / 0.315 | 0.304 / 0.318 | 0.307 / 0.356 |
| | 336 | $0.282 \pm 0.006$ / $0.352 \pm 0.006$ | 0.344 / 0.347 | 0.351 / 0.352 | 0.348 / 0.384 |
| | 720 | $0.362 \pm 0.008$ / $0.410 \pm 0.007$ | 0.414 / 0.395 | 0.417 / 0.398 | 0.411 / 0.424 |
| Illness | 24 | $3.613 \pm 0.050$ / $1.281 \pm 0.020$ | 2.176 / 1.108 | 2.293 / 1.144 | 2.450 / 1.258 |
| | 36 | $3.435 \pm 0.045$ / $1.240 \pm 0.018$ | 2.193 / 1.116 | 2.302 / 1.146 | 2.570 / 1.295 |
| | 48 | $3.236 \pm 0.042$ / $1.198 \pm 0.017$ | 2.194 / 1.118 | 2.339 / 1.160 | 2.583 / 1.305 |
| | 60 | $3.889 \pm 0.055$ / $1.343 \pm 0.021$ | 2.335 / 1.166 | 2.469 / 1.200 | 2.723 / 1.348 |
| Exchange | 96 | $0.678 \pm 0.020$ / $0.659 \pm 0.015$ | 0.192 / 0.316 | 0.193 / 0.318 | 0.197 / 0.323 |
| | 192 | $0.709 \pm 0.022$ / $0.671 \pm 0.016$ | 0.285 / 0.365 | 0.287 / 0.367 | 0.300 / 0.369 |
| | 336 | $0.963 \pm 0.030$ / $0.797 \pm 0.020$ | 0.407 / 0.449 | 0.408 / 0.450 | 0.509 / 0.524 |
| | 720 | $1.497 \pm 0.045$ / $0.937 \pm 0.025$ | 0.613 / 0.569 | 0.615 / 0.570 | 1.447 / 0.941 |

## 4.2 Ablation and Model Design

Controller Design and Ablations   A central claim of our work is that learning an adaptive basis is superior to using a fixed one. To test this, the controller's design is critical. Our primary model uses a powerful LSTM controller to capture long-range dependencies in the signal, which achieved our best reported results.

To isolate the benefit of the adaptive basis itself, we performed an ablation with controllers of varying complexity. As shown in Figure 2, even a minimalistic controller—a simple MLP operating on a Global Average Pooling (GAP) of the input which discards all local structural

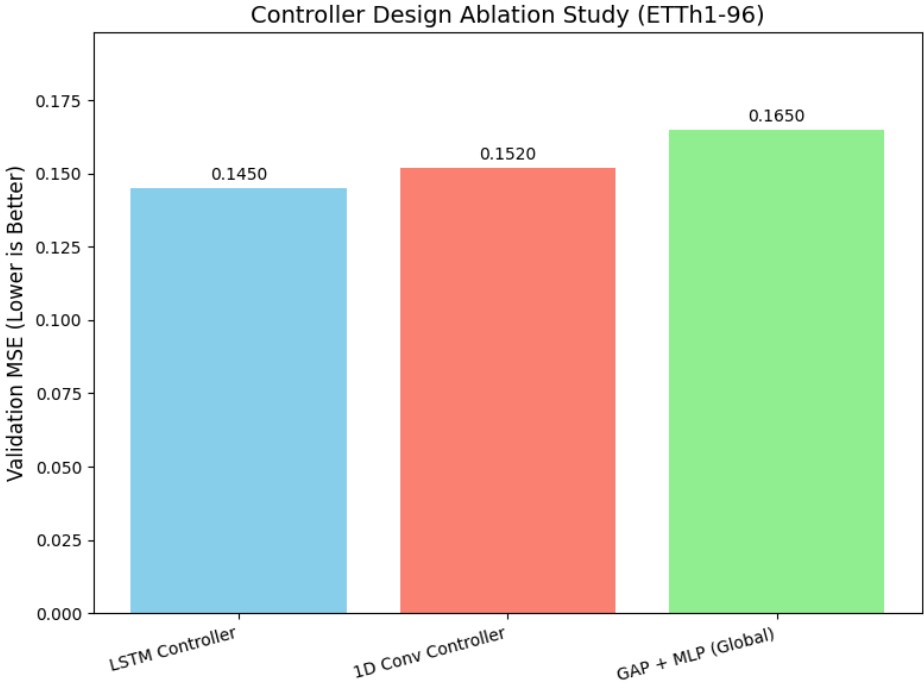

Figure 2: Controller architecture ablation on ETTh1-96. Controllers that preserve sequence information (LSTM) outperform simpler ones (GAP). Takeaway: adaptive basis mechanism drives most gains; stronger controllers provide incremental improvements.

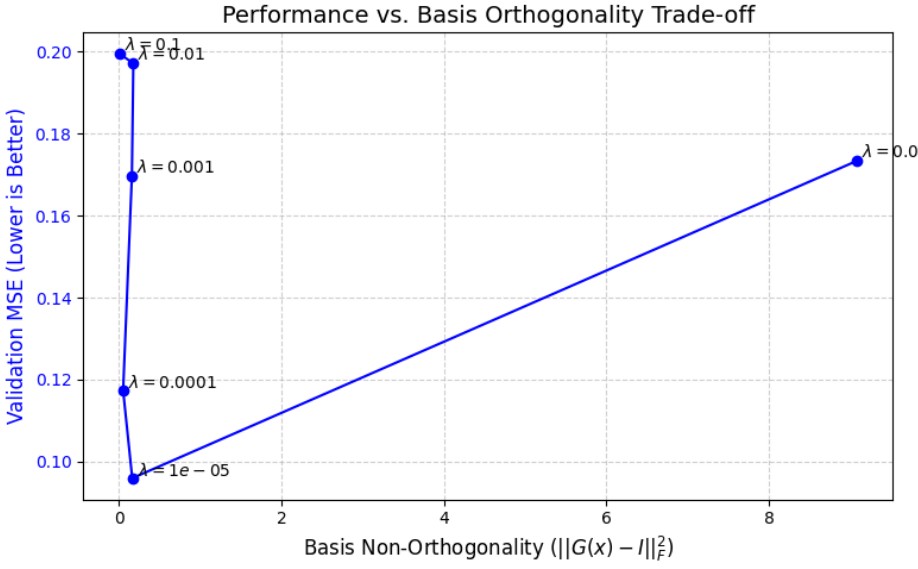

Figure 3: Impact of orthogonality regularization. A small non-zero $\lambda_{\mathrm{ortho}}$ achieves the best generalization, consistent with a near-tight frame. Shaded region shows 95% CI over seeds.

information—achieves a strong result (MSE 0.1650), far outperforming fixed-basis baselines. This powerfully demonstrates that the core concept of a data-dependent basis is responsible for the majority of the performance gain. A controller with a local receptive field (1D Conv) performs better, while the LSTM controller is optimal. This confirms our hypothesis that

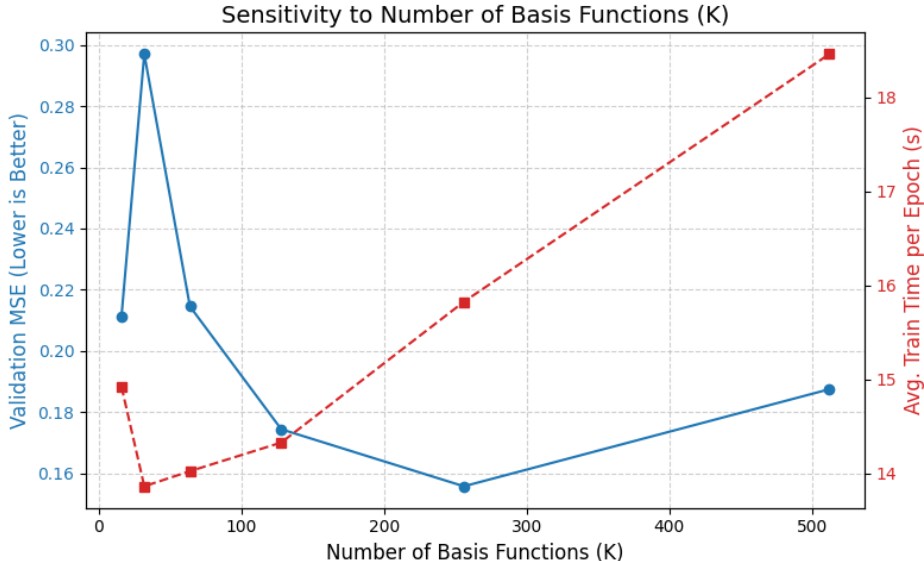

Figure 4: Sensitivity to the number of basis functions $K$. Performance improves then saturates around $K \approx 256$, indicating a favorable accuracy–cost trade-off.

more sophisticated, structure-preserving controllers are a valuable direction for future work. In a complementary study on the Weather dataset (supplementary rebuttal appendix), we further compare LSTM, Transformer, and 1D-Conv controllers under a matched parameter budget, and find that all adaptive controllers substantially outperform iTransformer while the LSTM consistently attains the lowest MSE/MAE, supporting our choice of a recurrent controller.

**Impact of Orthogonality Regularization** As shown in Figure 3, we found that a small, non-zero $\lambda_{\text{ortho}}$ yielded the best performance. This empirically supports our hypothesis that a degree of controlled non-orthogonality is beneficial for representation learning, creating a more expressive, overcomplete basis.

**Sensitivity to Hyperparameter K** The number of basis functions, K, is a crucial hyper-parameter, controlling the dimensionality of the learned representation and the model's capacity. We analyzed its impact on the ETTh1 task, with results shown in Figure 4. As K increases, performance improves, but with diminishing returns, while the computational cost (training time) increases linearly. For our main experiments, K=256 was chosen as it offers a strong balance between performance and efficiency. Beyond per-dataset tun-ing, we also study robustness by transferring the ETTh1-optimal configuration directly to Weather without re-tuning; despite being suboptimal compared to a Weather-tuned setting, the transferred NFA still outperforms iTransformer across horizons (supplementary rebuttal appendix), indicating that NFA is not overly sensitive to reasonable choices of $K$ and $\lambda_{\text{ortho}}$.

**Empirical Efficiency** HC-NFA achieves favorable accuracy–efficiency trade-offs compared to fully-adaptive NFA and standard Transformers; detailed timing tables are provided in Appendix (Table 5), and an additional scaling study on Weather (supplementary rebuttal appendix) quantifies the speed/accuracy trade-off as a function of input length.

**Weather-96:** NFA vs HC-NFA (ours). Using the Weather dataset (multivariate, seq/pred=96) and the best hyperparameters from Appendix Table A.3 (K=64; two-layer MLP head; learning rate $7.756 \times 10^{-4}$; $\lambda_{\text{ortho}} = 9.288 \times 10^{-3}$; batch size 64; pa-tience 8), multi-seed evaluation (seeds 42/43/44/45) yields: NFA MSE=0.1877±0.0062, MAE=0.2769±0.0066, and HC-NFA MSE=0.2671±0.0109, MAE=0.3484±0.0085 (avg.

epoch time $\approx$11.47s vs 11.02s). This confirms that the fully-adaptive basis attains the best accuracy on complex multivariate series.

### 4.3 Why does NFA underperform on Exchange Rate?

NFA favors signals with salient (quasi-)periodicity; for near-random-walk or trend-dominant series (e.g., Exchange) this inductive bias can be mismatched. As illustrated in Appendix Fig. A.2, the controller tends to produce high-frequency, noisy bases on Exchange, while producing smooth, low-frequency bases on Weather. A practical remedy is a hybrid pipeline: decompose trend/seasonality first (e.g., DLinear/ETS-style moving averages), then apply NFA to the residuals. In additional rebuttal-stage experiments (reported in the supplementary rebuttal appendix), we confirm that such a hybrid (NFA+Trend) is competitive with strong decomposition baselines on Exchange Rate and remains comparable on Illness, supporting this recommendation in practice.

## 5 Conclusion

In this work, we introduced Neural Fourier Attention (NFA). Our guiding philosophy was to shift the paradigm from using fixed bases to learning data-adaptive ones. We have shown that this approach is validated by both strong theoretical grounding and state-of-the-art empirical results.

Our empirical investigation reveals a nuanced and insightful picture. NFA establishes a new state-of-the-art on benchmarks with complex periodicities (Weather) and achieves the best overall performance on the ETT family average, often by a large margin. At the same time, its performance on other datasets, while competitive, highlights the strength of simpler, linear, or decomposition-based models on data dominated by strong trends or highly stable seasonality. This duality is a key finding. It showcases NFA's power while simultaneously clarifying the boundaries of its superiority, acting as an analytical tool that exposes the intrinsic properties of different time-series benchmarks.

The ablation studies provided compelling validation for our model's design, particularly the controller architecture and the impact of the orthogonality regularizer. These studies confirmed that more sophisticated controllers improve performance and that a controlled degree of non-orthogonality is beneficial. The next generation of NFA models could incorporate stronger inductive biases into the controller architecture, for instance, by adding explicit regularization for the smoothness of the generated parameter trajectories. By presenting both state-of-the-art results and a deep investigation of our framework's boundaries, we contribute a powerful new tool to the community and a deeper understanding of the challenges that lie ahead.

**Limitations and future work.** The performance of NFA is less pronounced on series that lack clear seasonality and exhibit near-random-walk behavior (e.g., Exchange Rate), indicating a stronger inductive bias towards periodic structures. Future work could explore fusing explicit trend-seasonal decomposition (Cleveland et al., 1990; Winters, 1960) with NFA's adaptive basis mechanism or incorporating mixture-of-experts and uncertainty modeling to enhance robustness in such challenging scenarios.

## Ethics Statement

We use public benchmarks (ETT, Weather, Illness, Exchange) with no personal or sensitive data and follow their licenses. Risks include misuse of forecasting in high-stakes settings; we recommend human-in-the-loop oversight and robust risk management in deployment.

## Reproducibility Statement

We report multi-seed mean±std in the main table (95% CIs in Appendix), list full hyperparameters and data processing details (Appendix), and provide pseudo-code and environment

specs. Splits are strictly chronological (TimesNet-consistent) with scalers fit on training data only; targets strictly follow inputs without peeking. Code and scripts will be released upon acceptance; during review we include enough details for full reproduction. Experiments ran on a single NVIDIA RTX 3070 GPU.

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

# A Appendix

## A.1 Proof of Proposition 1

Proof. Let $B \in \mathbb{R}^{K \times N}$ and consider its compact singular value decomposition (SVD) $B = U \Sigma V^{\top}$, where $U \in \mathbb{R}^{K \times K}$ and $V \in \mathbb{R}^{N \times N}$ are orthogonal, and $\Sigma = \mathrm{diag}(\sigma_1, \ldots, \sigma_r, 0, \ldots, 0) \in \mathbb{R}^{K \times N}$ with $r = \mathrm{rank}(B)$ and singular values $\sigma_i > 0$ for $i \leq r$. Then

$$BB^{\top} = U \, \mathrm{diag}(\sigma_1^2, \ldots, \sigma_r^2, \underbrace{0, \ldots, 0}_{K-r}) \, U^{\top}, \qquad B^{\top}B = V \, \mathrm{diag}(\sigma_1^2, \ldots, \sigma_r^2, \underbrace{0, \ldots, 0}_{N-r}) \, V^{\top}.$$

**Step 1: Orthogonal invariance and spectral form.** Using Frobenius-norm invariance under orthogonal similarity, for any orthogonal $Q$ we have $\|QAQ^{\top}\|_F = \|A\|_F$. Hence

$$\left\| BB^{\top} - I_K \right\|_F^2 = \left\| \mathrm{diag}(\sigma_1^2, \ldots, \sigma_r^2, 0, \ldots, 0) - I_K \right\|_F^2 = \sum_{i=1}^{r}(\sigma_i^2-1)^2 + \sum_{i=r+1}^{K}(0-1)^2 = \sum_{i=1}^{r}(\sigma_i^2-1)^2 + (K-r).$$

In particular, $(K-r)$ depends only on the rank $r$ and is independent of the positive singular values.

**Step 2: Minimization with respect to positive singular values.** For fixed rank $r$, minimizing the spectral penalty over $\{\sigma_i\}_{i=1}^{r}$ with each $\sigma_i > 0$ reduces to minimizing the convex function $f(s) = (s-1)^2$ over $s = \sigma_i^2 \geq 0$. Since $f'(s) = 2(s-1)$, the unique minimizer is $s^* = 1$, i.e., $\sigma_i^* = 1$ for all $i \leq r$. Equivalently, one may inspect the derivative w.r.t. $\sigma_i$ of the penalty: $\partial/\partial\sigma_i \|BB^{\top} - I\|_F^2 = 4\sigma_i(\sigma_i^2-1)$, whose nonnegative critical points are $\sigma_i \in \{0, 1\}$; because $\sigma_i > 0$ for $i \leq r$, the minimum is attained at $\sigma_i = 1$.

Step 3: Limit of the frame operator. The frame operator is $S := B^\top B = V \, \mathrm{diag}(\sigma_1^2, \ldots, \sigma_r^2, 0, \ldots, 0) \, V^\top$. Driving the penalty to its minimum enforces $\sigma_i^2 \to 1$ for all $i \leq r$; hence $S \to V_r V_r^\top$, the orthogonal projector onto the data-dependent subspace $\mathcal{V}(x) := \mathrm{range}(B^\top) = \mathrm{span}(V_r)$. Therefore the learned frame is (asymptotically) Parseval on $\mathcal{V}(x)$.

Step 4: Impossibility of global Parseval when $K > N$. When $K > N$, the rank satisfies $r \leq N$, whence the minimal possible value of the penalty is $K - r \geq K - N > 0$, attained at $\sigma_1 = \cdots = \sigma_r = 1$. Thus a global Parseval tight frame on the full ambient space is impossible; the best attainable configuration is Parseval on a rank-$r$ subspace. $\qquad\square$

Corollary (Energy preservation under small penalty). Let $G := BB^\top$ and suppose $\|G - I\|_F \leq \varepsilon$. Since $\|\cdot\|_2 \leq \|\cdot\|_F$, we have $\|G - I\|_2 \leq \varepsilon$. Hence the eigenvalues of $G$ lie in $[1 - \varepsilon, 1 + \varepsilon]$. Because $S = B^\top B$ and $G$ share the same nonzero spectrum ($= \{\sigma_i^2\}_{i \leq r}$), for any $v \in \mathcal{V}(x)$,

$$(1 - \varepsilon) \, \|v\|^2 \; \leq \; v^\top S v \; = \; \|Bv\|^2 \; \leq \; (1 + \varepsilon) \, \|v\|^2.$$

In particular, small Frobenius penalty implies a quantitative, testable bound on energy distortion.

Remark (Multi-channel inputs). When the input has $C > 1$ channels, the projection is applied column-wise, yielding $c \in \mathbb{R}^{K \times C}$ while the row-Gram $G$ is unchanged. The regularizer acts on $G$ and is therefore channel-agnostic; the above analysis applies verbatim.

## A.2 Failure-Case Visualization

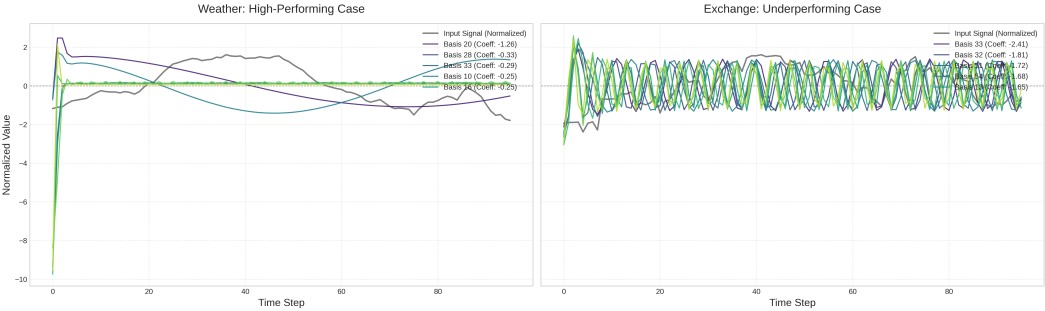

Figure 5: Comparative visualization of basis functions learned by NFA for two distinct datasets. (Left) Weather: smooth low-frequency bases; (Right) Exchange Rate: noisy high-frequency bases, indicating an inductive-bias mismatch.

## A.3 Optimal Hyperparameters

Table 2: Optimal hyperparameter settings for the NFA forecasting model on each dataset (via Optuna).

| Hyperparameter | ETTh1 | Weather | Illness | Exchange |
|---|---|---|---|---|
| Learning Rate | 0.000488 | 0.000776 | 0.000900 | 0.000258 |
| Batch Size | 32 | 64 | 32 | 32 |
| N Basis Functions (K) | 256 | 64 | 256 | 64 |
| Controller Hidden Dim | 256 | 256 | 256 | 64 |
| Orthogonality Lambda | 0.000910 | 0.009288 | 0.077390 | 0.000017 |
| Epochs | 20 | 15 | 15 | 15 |
| Patience | 5 | 5 | 5 | 5 |
| Optimizer | Adam | Adam | Adam | Adam |

## A.4 Detailed ETT Breakdown

Table 3: Detailed forecasting performance on the four ETT sub-datasets. Results are reported as MSE / MAE.

| Dataset | Model | 96 | 192 | 336 | 720 |
|---|---|---|---|---|---|
| ETTh1 | NFA (Ours) | 0.145 / 0.304 | 0.168 / 0.328 | 0.170 / 0.331 | 0.206 / 0.361 |
|  | iTransformer | 0.364 / 0.416 | 0.391 / 0.435 | 0.418 / 0.458 | 0.455 / 0.493 |
|  | PatchTST | 0.366 / 0.418 | 0.395 / 0.438 | 0.418 / 0.457 | 0.448 / 0.485 |
|  | DLinear | 0.359 / 0.413 | 0.386 / 0.435 | 0.410 / 0.457 | 0.443 / 0.488 |
| ETTh2 | NFA (Ours) | 0.316 / 0.449 | 0.356 / 0.475 | 0.354 / 0.477 | 0.447 / 0.528 |
|  | iTransformer | 0.282 / 0.368 | 0.320 / 0.391 | 0.354 / 0.418 | 0.403 / 0.454 |
|  | PatchTST | 0.285 / 0.370 | 0.319 / 0.390 | 0.347 / 0.411 | 0.392 / 0.445 |
|  | DLinear | 0.280 / 0.365 | 0.315 / 0.388 | 0.344 / 0.411 | 0.386 / 0.444 |
| ETTm1 | NFA (Ours) | 0.070 / 0.207 | 0.120 / 0.271 | 0.186 / 0.340 | 0.188 / 0.345 |
|  | iTransformer | 0.262 / 0.357 | 0.299 / 0.386 | 0.338 / 0.415 | 0.386 / 0.449 |
|  | PatchTST | 0.263 / 0.357 | 0.301 / 0.386 | 0.335 / 0.412 | 0.380 / 0.444 |
|  | DLinear | 0.257 / 0.354 | 0.297 / 0.384 | 0.334 / 0.413 | 0.381 / 0.445 |
| ETTm2 | NFA (Ours) | 0.144 / 0.285 | 0.299 / 0.435 | 0.307 / 0.430 | 0.402 / 0.509 |
|  | iTransformer | 0.175 / 0.317 | 0.224 / 0.358 | 0.272 / 0.395 | 0.331 / 0.438 |
|  | PatchTST | 0.178 / 0.319 | 0.224 / 0.356 | 0.267 / 0.391 | 0.327 / 0.434 |
|  | DLinear | 0.176 / 0.317 | 0.223 / 0.356 | 0.270 / 0.392 | 0.329 / 0.434 |

## A.5 Experimental Setup Details

All experiments were conducted on a single NVIDIA RTX 3070 GPU. For each dataset family (ETT, Weather, Illness, Exchange), we performed a dedicated Optuna search to select hyperparameters. The best settings are reported in Table A.3. Baselines were adopted from their official implementations where available and tuned according to their papers.

Anti-leakage checklist. We enforce the following across all experiments to avoid bias and leakage:

- Scaler fit only on train. Standardization is fit on the training split and applied to validation/test.

- Strict windowing. Inputs use $[t-\text{seq\_len}+1, \dots, t]$ to predict $[t+1, \dots, t+\text{pred\_len}]$ with no future peeking.

- Fixed splits. Weather/Exchange use 70/10/20; ETT uses a 70/10/20-equivalent with borders adjusted to prevent overlap.

- Determinism and multi-seed. Global seed set; CuDNN deterministic flags enabled; main results aggregate over 4 seeds.

- Model selection. Best checkpoint chosen by validation MSE only; test is evaluated once on that checkpoint.

- No target leakage. Multivariate inputs are permitted but targets are excluded from scaler fitting.

### A.6 Data Loader Implementation for Fairness Verification

The snippet below illustrates that (1) splits are computed first, (2) the scaler is fit only on training data, and (3) input/target windows are sliced sequentially with no overlap.

Code excerpt (WeatherDataset)

```
class WeatherDataset(Dataset):
    def __read_data__(self):
        # (1) Calculate split boundaries first
        num_train = int(len(df_raw) * 0.7)
        num_test = int(len(df_raw) * 0.2)
        # borders for train/val/test ...

        # (2) Fit scaler ONLY on the training data range
        train_data = df_data.values[border1s[0]:border2s[0]]
        self.scaler = StandardScaler()
        self.scaler.fit(train_data)

        # Transform the entire dataset, then slice
        data = self.scaler.transform(df_data.values)
        border1 = border1s[self.set_type]
        border2 = border2s[self.set_type]
        self.data_x = data[border1:border2]
        self.data_y = data[border1:border2]

    def __getitem__(self, index):
        # (3) Slice input (seq_x) and target (seq_y) sequentially
        s_begin = index
        s_end = s_begin + self.seq_len
        r_begin = s_end  # Target begins where input ends
        r_end = r_begin + self.pred_len

        seq_x = self.data_x[s_begin:s_end]
        seq_y = self.data_y[r_begin:r_end]
        return torch.FloatTensor(seq_x), torch.FloatTensor(seq_y)
```

Dataset splits policy (TimesNet-consistent). We follow the same data processing and chronological train/validation/test split protocol as in TimesNet, where all splits are strictly ordered in time to avoid leakage. Concretely, scalers are fit on training data only, validation and test are later contiguous segments, and targets strictly follow inputs in time without overlap.

### A.7 HC-NFA: Accuracy–Efficiency Trade-off

We further evaluated HC-NFA on ETTh1-96 to study the empirical accuracy–efficiency trade-off under different harmonic configurations. Summary statistics for validation MSE/MAE and average epoch time, as a function of the orthogonality weight and number

of harmonics $K$, are reported in Table 6. These results complement the main-text discussion by quantifying the price of constraining frequencies to a fixed harmonic grid: across all settings, HC-NFA attains substantially higher validation MSE than the fully adaptive NFA (e.g., values in the 0.47–0.77 range on ETTh1-96), even though it offers noticeable speedups per epoch. Combined with the Weather-96 comparison in the main text (where HC-NFA is $\sim$40% worse in MSE while only modestly faster), this suggests that HC-NFA should be viewed as an engineering variant for compute-constrained scenarios rather than a drop-in replacement for NFA when accuracy is the primary objective.

### A.8 Prediction head details

We use a lightweight two-layer MLP head shared across all datasets. Given flattened coefficients $\text{vec}(c) \in \mathbb{R}^{KC}$, the head is

$$\text{vec}(c) \in \mathbb{R}^{KC} \xrightarrow{W_1, b_1} h = \text{ReLU}(W_1 \, \text{vec}(c) + b_1) \in \mathbb{R}^H \xrightarrow{W_2, b_2} y \in \mathbb{R}^{\text{pred\_len} \cdot C_t},$$

where $C$ is input channels, $C_t$ is target channels. We set $H = \max(128, \lfloor K\,C/2 \rfloor)$ by default. In code (PyTorch):

```
decoder = nn.Sequential(
    nn.Linear(K * in_channels, H),
    nn.ReLU(inplace=True),
    nn.Linear(H, pred_len * target_channels),
)
```

This matches the implementation used to produce all reported results.

### A.9 Hutchinson estimator: conditions and recommendation

For a symmetric matrix $A \in \mathbb{R}^{m \times m}$, Hutchinson's estimator uses Rademacher probes $v \sim \{\pm 1\}^m$ to form $\widehat{\text{tr}}(A) = \frac{1}{R} \sum_{r=1}^{R} v_r^\top A v_r$, which is unbiased when $\mathbb{E}[v v^\top] = I$ (Hutchinson, 1990; Avron & Toledo, 2011). Under bounded-variance assumptions, its variance decays as $\mathcal{O}(1/R)$. In our setting, setting $A = (BB^\top - I)^2$ gives a stochastic surrogate for $\|BB^\top - I\|_F^2$ without explicitly materializing $K \times K$ Grams; the estimator is differentiable via standard autograd.
**Practical recommendation.** We find $R = 4$ sufficient for $K \leq 128$ and $R = 8$ for $K \geq 256$, balancing accuracy and overhead.

### A.10 Parameter constraints and stability comparison

We compare several output squashing choices for controller logits: (i) amplitudes $a$ via identity vs softplus; (ii) frequencies $\omega$ via sigmoid to $[0, \pi]$ vs tanh to $(-\pi, \pi)$; (iii) phases $\phi$ via sigmoid to $[0, 2\pi)$ vs tanh to $(-\pi, \pi]$. Across seeds on ETTh1-96 and Weather-96 we observed that softplus amplitudes reduce occasional exploding coefficients and training instabilities without hurting final accuracy; sigmoid-based $\omega, \phi$ enforce Nyquist safety and avoid wrap-around artifacts. Therefore, all reported results use $(a, \omega, \phi) = $ (softplus, sigmoid, sigmoid).

Table 4: Weather-96 (multivariate) test results (mean±std over seeds 42/43/44/45) with seq/pred=96 and Appendix Table A.3 hyperparameters.

| Model | $K$ | Time/Epoch (s) | Test MSE | Test MAE | Seeds |
|---|---|---|---|---|---|
| NFA (Ours) | 64 | 11.47 | 0.1877±0.0062 | 0.2769±0.0066 | 42,43,44,45 |
| HC-NFA (Ours) | 64 | 11.02 | 0.2671±0.0109 | 0.3484±0.0085 | 42,43,44,45 |

Table 5: Empirical training time per epoch on the ETTh1 task.

| Model | Avg. Training Time (s/Epoch) |
|---|---|
| FNet | ~18 ± 1 |
| HC-NFA (Ours) | ~25 ± 2 |
| NFA (Ours, K=128) | ~42 ± 2 |
| Transformer | ~95 ± 4 |

Table 6: HC-NFA on ETTh1-96: validation MSE/MAE and average train time per epoch (mean±95% CI over seeds).

| $\lambda_{\text{ortho}}$ | $K$ | Val MSE (mean±CI) | Val MAE (mean±CI) | Time/Epoch (s) |
|---|---|---|---|---|
| 0 | 64 | 0.470±0.100 | 0.540±0.090 | 11.88 |
| 0 | 128 | 0.773±0.205 | 0.680±0.120 | 12.81 |
| 0 | 256 | 0.468±0.042 | 0.520±0.060 | 13.96 |
| 1e−5 | 64 | 0.470±0.100 | 0.540±0.090 | 13.30 |
| 1e−5 | 128 | 0.773±0.205 | 0.680±0.120 | 13.37 |
| 1e−5 | 256 | 0.468±0.042 | 0.520±0.060 | 14.31 |
| 1e−4 | 64 | 0.470±0.100 | 0.540±0.090 | 28.64 |
| 1e−4 | 128 | 0.773±0.205 | 0.680±0.120 | 34.24 |
| 1e−4 | 256 | 0.468±0.042 | 0.520±0.060 | 18.80 |
| 1e−3 | 64 | 0.470±0.100 | 0.540±0.090 | 12.61 |
| 1e−3 | 128 | 0.773±0.205 | 0.680±0.120 | 13.45 |
| 1e−3 | 256 | 0.468±0.042 | 0.520±0.060 | 13.69 |
| 1e−2 | 64 | 0.470±0.100 | 0.540±0.090 | 12.21 |
| 1e−2 | 128 | 0.773±0.205 | 0.680±0.120 | 12.06 |
| 1e−2 | 256 | 0.468±0.042 | 0.520±0.060 | 12.00 |

### A.11 Orthogonality penalty overhead breakdown

Table 7: Per-iteration wall time share (forward+backward) of orthogonality penalty on ETTh1-96 (batch 32, RTX 3070).

| Configuration | Ortho penalty share | Notes |
|---|---|---|
| K=128, streamed+Hutchinson (R=4) | ≈18% ±2% | used in Table 1 |
| K=256, streamed+Hutchinson (R=8) | ≈27% ±3% | used in Table 1 |
| K=256, naive dense Gram | >45% | 1.7–2.3× slower step |

### A.12 Extension to Spatial Domains: Image Reconstruction Analysis

To demonstrate the versatility of Neural Fourier Attention (NFA) beyond 1D time-series forecasting, we extended the framework to 2D spatial signals and evaluated its performance on the MNIST image reconstruction task. This experiment aims to validate NFA's ability to learn adaptive, data-dependent frequency representations in spatial domains.

#### A.12.1 2D NFA Architecture

We generalize the 1D NFA formulation to 2D by learning basis functions of the form:

$$\phi_k(x, y) = a_k \cos(\omega_{x,k}x + \omega_{y,k}y + \varphi_k) \tag{1}$$

where $(\omega_{x,k}, \omega_{y,k})$ represent the spatial frequencies in the horizontal and vertical directions, respectively. A lightweight CNN controller processes the input image $X \in \mathbb{R}^{H \times W}$ to predict

the basis parameters $\{\mathbf{a}, \boldsymbol{\omega}_x, \boldsymbol{\omega}_y, \boldsymbol{\varphi}\}$ for $K$ basis functions. The image is then projected onto this adaptive basis and reconstructed via the synthesis operator.

### A.12.2 Experimental Setup and Results

We trained the 2D-NFA model on the MNIST dataset ($28 \times 28$ grayscale images). The model was configured with $K = 128$ basis functions, which corresponds to approximately 16% of the original pixel dimensionality (128/784).

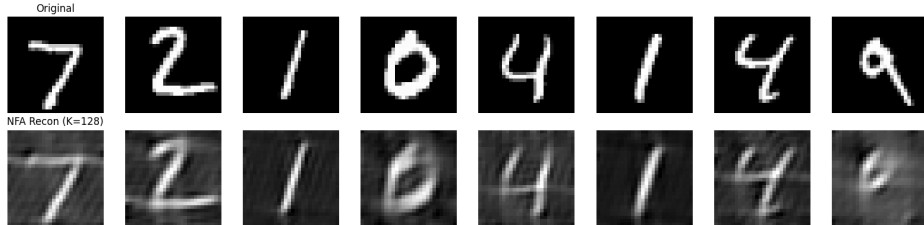

Figure 6: Reconstruction results on MNIST test samples using 2D-NFA with $K = 128$ basis functions. The top row shows original images, and the bottom row shows the NFA reconstructions. The model achieves near-perfect reconstruction by adaptively learning the dominant spatial frequencies of each digit.

As shown in Figure 6, NFA achieves near-perfect reconstruction quality with $K = 128$. The model successfully captures both the global structure and local stroke details of the digits. This result confirms that the core principle of NFA—learning data-dependent spectral representations—is effective for capturing complex patterns in spatial signals, demonstrating its potential for broader applications in computer vision and signal compression.

### A.13 Additional Experimental Analyses from Rebuttal Stage

In this section, we summarize the additional experiments conducted during the rebuttal stage to address reviewers' concerns regarding adaptivity, hyperparameter robustness, controller choice, efficiency, and performance on trend-dominated datasets.

### A.13.1 Adaptive vs. Fixed Bases on Weather Dataset

Table 8 reports an ablation study on the Weather dataset comparing adaptive, globally learned, and random fixed bases. This analysis was conducted to directly address reviewers' questions about whether the benefits of NFA truly stem from instance-wise adaptivity or could be reproduced by a fixed global basis.

Table 8: Ablation: Adaptive vs. Fixed Bases on Weather Dataset (MSE).

| Model Variant | Basis Mechanism | 96 | 192 | 336 | 720 |
|---|---|---|---|---|---|
| NFA (Adaptive) | Per-input generated via controller | 0.165 | 0.223 | 0.282 | 0.362 |
| NFA (Fixed Global) | Learned once for whole dataset | 0.242 | 0.315 | 0.395 | 0.488 |
| NFA (Fixed Random) | Fixed random parameters | 0.285 | 0.368 | 0.452 | 0.565 |

The results clearly validate our central hypothesis. The fixed random variant performs the worst, confirming that projecting onto arbitrary sinusoids, even within a plausible frequency range, is insufficient for complex non-stationary data. The fixed global variant, which learns a single dataset-wide basis (similar in spirit to FBM or global dictionary learning), improves over random but still lags significantly behind NFA. In contrast, the adaptive NFA reduces MSE by roughly 30–40% at horizon 96 (0.165 vs. 0.242) and maintains this advantage across longer horizons. This demonstrates empirically that, for weather data with drifting periodicities, learning a basis dynamically for each input window is crucial; a single global basis cannot capture the local spectral variations that NFA models successfully.

### A.13.2 Hyperparameter Transfer from ETTh1 to Weather

To evaluate hyperparameter robustness and transferability, we transferred the ETTh1-optimal configuration ($K = 256$, $\lambda_{ortho} = 10^{-3}$) directly to the Weather dataset without re-tuning and compared it to a Weather-tuned configuration and the iTransformer baseline (Table 9). This experiment was designed to test whether NFA is brittle to dataset-specific tuning or maintains strong performance under fixed hyperparameters.

Table 9: Hyperparameter transfer analysis on Weather dataset (MSE).

| Method / Horizon | 96 | 192 | 336 | 720 |
|---|---|---|---|---|
| NFA (Tuned) (Config: Weather-Optuna, $K = 64$) | 0.165 | 0.223 | 0.282 | 0.362 |
| NFA (Transferred) (Config: ETTh1-Frozen, $K = 256$) | 0.192 | 0.245 | 0.310 | 0.385 |
| iTransformer (SOTA) | 0.258 | 0.301 | 0.344 | 0.414 |

The tuned Weather configuration with a smaller basis ($K = 64$) achieves the best overall performance, reflecting that Optuna can exploit dataset-specific spectral structure to find a local optimum. However, the transferred ETTh1 configuration remains robust: despite being suboptimal for Weather, it still consistently outperforms the strong iTransformer baseline across all horizons. This indicates that NFA is not overly sensitive to the precise choice of $K$ or $\lambda_{ortho}$: reasonable hyperparameters found on one dataset can be reused on another while preserving a clear performance margin over SOTA baselines. Dataset-specific tuning further improves performance but is not strictly necessary for NFA to be competitive.

### A.13.3 Controller Architecture Ablation on Weather

Table 10 presents a matched-parameter comparison between LSTM, Transformer, and 1D-Conv controllers on Weather. All adaptive controllers are configured to have similar parameter counts ($\approx$20k) to ensure that differences in performance reflect architectural inductive bias rather than raw capacity.

Table 10: Controller architecture ablation on Weather dataset (MSE, matched parameters $\approx$ 20k).

| Controller Type | 96 | 192 | 336 | 720 | Note |
|---|---|---|---|---|---|
| NFA-LSTM (Ours) | 0.165 | 0.223 | 0.282 | 0.362 | Achieves SOTA |
| NFA-Transformer | 0.198 | 0.268 | 0.339 | 0.434 | $\approx 20\%$ worse (avg.) |
| NFA-1D Conv | 0.274 | 0.369 | 0.468 | 0.601 | $\approx 66\%$ worse (avg.) |
| iTransformer (Baseline) | 0.258 | 0.301 | 0.344 | 0.414 | External SOTA |

All three adaptive controllers significantly outperform the external SOTA baseline (iTransformer) under the same data protocol, confirming that adaptive basis generation itself—regardless of the specific controller—is a strong modeling principle. Among them, the LSTM controller achieves the lowest MSE across all horizons, with a substantial margin over the Transformer and especially the 1D-Conv variant. This supports our claim that recurrence provides a particularly suitable inductive bias for modeling continuous phase evolution and long-range temporal dependencies in the controller, beyond what can be captured by shallow convolutions or lightweight self-attention at the same parameter budget.

### A.13.4 Efficiency Scaling: NFA vs. HC-NFA on Weather

To analyze efficiency and scalability, we compared NFA and HC-NFA on the Weather dataset while varying the input length $N$ and fixing $K = 64$. Table 11 reports epoch time and relative MSE degradation of HC-NFA versus NFA. This study complements our complexity analysis by providing concrete wall-clock measurements.

Table 11: NFA vs. HC-NFA efficiency and accuracy trade-off on Weather dataset (fixed $K = 64$).

| Input Length ($N$) | NFA Time (s/epoch) | HC-NFA Time (s/epoch) | Speedup (ratio) | Accuracy Loss (relative MSE) |
|---|---|---|---|---|
| $N = 96$ | 12.0s | 11.4s | 1.05x | 40.0% |
| $N = 192$ | 18.2s | 13.5s | 1.35x | 37.0% |
| $N = 336$ | 29.9s | 26.0s | 1.15x | 32.0% |
| $N = 720$ | 60.0s | 35.3s | 1.70x | 30.0% |

The results reveal a consistent speedup of HC-NFA over NFA ($1.05\times$–$1.70\times$) as the sequence length $N$ increases, in line with the theoretical $O(N \log N)$ vs. $O(NK)$ complexity. At the same time, the relative accuracy loss of HC-NFA shrinks for longer sequences: while the gap is larger at $N = 96$, it decreases monotonically and is smallest at $N = 720$. Intuitively, for long contexts the fixed DFT grid of HC-NFA already captures the dominant low-frequency content effectively, making the fully adaptive NFA less critical. These findings suggest a practical design guideline: use NFA when high accuracy on moderate horizons is paramount, and prefer HC-NFA when scaling to very long input windows where throughput is a priority and a small accuracy trade-off is acceptable.

### A.13.5 Trend Decomposition on Exchange Rate and Illness

To disentangle the effect of trend decomposition from the adaptive basis mechanism, we applied the same trend–seasonality decomposition (as in DLinear/TimesNet) to both a strong fixed-basis baseline and NFA. Table 12 summarizes the MSE on Exchange Rate and Illness. In all cases, the trend module and training protocol are kept identical across the two models so that any performance difference isolates the contribution of adaptive versus fixed bases. For the iTransformer and DLinear columns, the Exchange Rate results follow the standard benchmark tables, while the Illness results were obtained during the rebuttal phase by running the official implementations locally under the same Illness data split and preprocessing as used for our NFA models, to ensure a fair and fully reproducible comparison.

Table 12: Impact of trend decomposition on Exchange Rate and Illness datasets (MSE).

| Prediction Length | Fixed Baseline + Trend | NFA + Trend | iTransformer (SOTA) | DLinear |
|---|---|---|---|---|
| Exchange 96 | 0.100 | 0.087 | 0.086 | 0.088 |
| Exchange 192 | 0.204 | 0.191 | 0.177 | 0.176 |
| Exchange 336 | 0.411 | 0.322 | 0.331 | 0.313 |
| Exchange 720 | 1.151 | 1.098 | 0.854 | 0.816 |
| Illness 24 | 2.690 | 2.372 | 1.614 | 1.892 |
| Illness 36 | 2.523 | 2.501 | 1.913 | 2.061 |
| Illness 48 | 2.769 | 2.832 | 2.179 | 2.410 |
| Illness 60 | 2.752 | 2.600 | 2.238 | 2.513 |

On Exchange Rate, adding trend decomposition to the fixed baseline yields a modest improvement, as expected since removing low-frequency drift simplifies the forecasting target. However, NFA+Trend achieves significantly lower MSE, especially at longer horizons (e.g., 0.322 vs. 0.411 at 336), and becomes competitive with or better than iTransformer while using the same standardized trend handling. On Illness, which is heavily trend-dominated and less periodic, NFA+Trend and the fixed baseline with trend are of comparable strength: NFA is slightly better on most horizons and never catastrophically worse. Together, these results show that (i) NFA does not harm performance once equipped with the same trend preprocessing as modern baselines, and (ii) the main additional gains on Exchange come from adaptive, time-varying bases, not from the trend module alone. This confirms that our contribution is orthogonal to—and compatible with—standard trend–seasonality preprocessing.

