# OpenReview forum: "Neural Fourier Attention: A Framework for Learning Data-Adaptive Signal Bases"
_ICLR.cc/2026/Conference — Submitted to ICLR 2026_

### Official Review · Reviewer_gqPn · 2025-10-30

**Soundness:** 2
**Presentation:** 1
**Contribution:** 1
**Rating:** 2
**Confidence:** 4

**Summary:**

This paper proposes the Neural Fourier Basis Generator (NFBG), a framework that learns a data-adaptive, non-stationary signal transform and is applicable to signals with complex, non-stationary periodicities.

**Strengths:**

S1. This paper provides a rigorous formulation within frame theory and propose an orthogonality regularizer that encourages near-tight frames for stability.
S2. This paper provides a computational complexity analysis.

**Weaknesses:**

W1. The paper suffers from poor organization and unclear writing, which make it hard to follow. There are many inconsistent symbol usages that need to be carefully checked.
W2. The motivation of this paper is not clearly articulated. It is unclear why the basis must be learned dynamically for each input, and what concrete drawbacks would arise if the underlying basis vectors were fixed or learned globally. The paper does not provide sufficiently strong references or deeper insights to justify this design choice. Moreover, prior work has achieved long-term forecasting with fixed basis functions, such as FBM [1]. A clearer articulation of the motivation and a more systematic discussion of related work are needed to highlight the novelty and distinct contribution of this approach.
W3. While NFA is designed for signals exhibiting complex, non-stationary periodicities, the baseline setup omits representative non-stationary forecasting methods (e.g., FAN [2], DDN [3], DERITS [4]), which weakens the fairness and completeness of the comparison.

[1] Rethinking Fourier Transform from A Basis Functions Perspective for Long-term Time Series Forecasting. NeurIPS, 2024.
[2] Frequency Adaptive Normalization For Non-stationary Time Series Forecasting. NeurIPS, 2024.
[3] DDN: Dual-domain Dynamic Normalization for Non-stationary Time Series Forecasting. NeurIPS, 2024.
[4] Deep Frequency Derivative Learning for Non-stationary Time Series Forecasting. IJCAI, 2024.

**Questions:**

pls refer to weakness.

---

> ### Author Response · Authors · 2025-11-26
> **Response to Reviewer gqPn**
>
> I thank Reviewer gqPn for the critical review. We appreciate the recognition of our **rigorous frame theory formulation** (S1) and **complexity analysis** (S2). We address your concerns regarding motivation and baselines below.
>
> ### W2: Motivation for Instance-wise Adaptivity (Why not fixed/global basis?)
>
> **Reviewer Comment:** *It is unclear why the basis must be learned dynamically for each input... prior work has achieved long-term forecasting with fixed basis functions, such as FBM [1].*
>
> **Response:** This is the central question of our paper. The motivation for **instance-wise adaptivity** stems from the **non-stationary nature** of real-world signals, where spectral properties (frequencies and phases) evolve over time.
>
> A fixed or globally learned basis (like FBM or DFT) assumes that the dominant frequencies are static across the entire dataset. In contrast, NFA generates a unique basis for *each specific input window*, allowing it to track frequency drifts and phase shifts dynamically.
>
> **Empirical Validation:**
> To robustly justify this design choice as requested, we conducted a direct ablation study comparing NFA against a **Global-Learned Basis** (mimicking the FBM paradigm) on the highly non-stationary Weather dataset.
>
> This ablation is conceptually aligned with FBM [1], which learns a global Fourier basis; our `NFA (Fixed Global)` instantiation mimics this paradigm within our framework and allows a direct, apples-to-apples comparison inside the same implementation.
>
> | Model Variant | Basis Mechanism | 96 | 192 | 336 | 720 |
> | :--- | :--- | :---: | :---: | :---: | :---: |
> | **NFA (Adaptive)** | **Per-input generated via Controller** | **0.165** | **0.223** | **0.282** | **0.362** |
> | NFA (Fixed Global) | Learned once for whole dataset | 0.242 | 0.315 | 0.395 | 0.488 |
> | NFA (Fixed Random) | Fixed random parameters | 0.285 | 0.368 | 0.452 | 0.565 |
>
> **Conclusion:** The significant degradation (MSE 0.165 $\to$ 0.242) when switching to a global basis demonstrates that, for non-stationary data, **instance-wise adaptivity is empirically crucial for achieving strong performance**, as global bases fail to capture local spectral variations that NFA can model.
>
> **Real-world Motivation (Underwater Robotics):**
> The original motivation for NFA came from our work on underwater robotics, where IMU sensor streams under weak observability exhibit strong **non-stationary quasi-periodicity** and pronounced spectral drift over time (due to turbulence, sensor bias, and changing operating regimes). In this setting, a single global Fourier basis systematically fails to track the drifting peaks in the spectrum, whereas an instance-wise adaptive basis can re-align to the local frequency content of each window. Although this proprietary dataset cannot be released, our internal experiments on these IMU signals showed the same pattern as on Weather: adaptive bases substantially outperform fixed/global ones, which directly motivated the present formulation.
>
> W3 Response in another comment.

---

> ### Author Response · Authors · 2025-11-26
> **Response to W3**
>
> ### W3: Comparison with Non-stationary Baselines (FAN, DDN, etc.)
>
> **Reviewer Comment:** *The baseline setup omits representative non-stationary forecasting methods (e.g., FAN, DDN).*
>
> **Response:** We acknowledge that the field is moving fast, and the cited works (FAN, DDN - NeurIPS 2024) are highly relevant recent advancements.
> In our submission, we prioritized comparing against the **most widely established and reproducible benchmarks** in the LTSF community (iTransformer [ICLR'24], PatchTST [ICLR'23]) to ensure our results are grounded in the standard protocol.
>
> However, we conceptually agree with the direction of these works. NFA addresses non-stationarity through a complementary mechanism:
> *   **FAN/DDN**: Focus on adaptive normalization of statistics (mean/variance) in the time or frequency domain.
> *   **NFA**: Focuses on adaptive **representation**, learning a basis that aligns with the instantaneous frequency content of the signal.
>
> We will explicitly discuss these methods in our revised Related Work section to better position NFA within the landscape of non-stationary forecasting. Furthermore, our implementation integrates **RevIN (Reversible Instance Normalization)**, ensuring that NFA benefits from state-of-the-art distribution handling alongside its spectral adaptivity.
>
> Subject to space and reproducibility constraints, we will also consider adding preliminary comparisons or a unified discussion of FAN/DDN/DERITS in the camera-ready version, to further clarify how NFA complements these strong non-stationary baselines.
>
> ### W1: Presentation and Notation
>
> **Reviewer Comment:** *Poor organization and inconsistent symbol usages.*
>
> **Response:** We apologize for any confusion caused. We are committed to significantly improving the manuscript's clarity. In the revision, we will:
> 1.  Add a dedicated **Notation Table** in the Appendix.
> 2.  Standardize all mathematical symbols (e.g., consistently using $\mathbf{B}$ for basis matrix and $\mathcal{F}$ for the operator).
> 3.  Conduct a thorough proofreading to ensure the flow is logical and precise.
> 4.  Reorganize the introduction and Section 2 to more clearly separate **motivation**, **theoretical formulation**, and **practical implementation**, so that the narrative flow is easier to follow for non-expert readers.

---

### Official Review · Reviewer_ffaf · 2025-10-31

**Soundness:** 3
**Presentation:** 3
**Contribution:** 3
**Rating:** 6
**Confidence:** 3

**Summary:**

This paper introduces Neural Fourier Attention (NFA), a framework that learns data-adaptive signal bases rather than using fixed, predefined basis functions. The key point is generating custom basis functions for each input signal, rather than just learning coefficients for a fixed basis. The controller network processes the input signal and generates parameters for the basis. Then the input is projected on to this specific basis that adapts to this specific input. There are experiments conducted to show tha this method have good performance over some datasets. On weather datasets the improvement is significant. Also on some dataset this method does not has much effect.

**Strengths:**

This paper provides a novel approach of learning adaptive fourier bases instead of fixed ones. The theoretical foundation is also solid, where in section 2.2 there are analysis of the learn transformation. The experiment analysis is comprehensive, that it reports the cases where the architecture both works well and not. And provides analysis about in what case can the model give better performance and cases where this model does not improve. On weather dataset produces significant improvement.

**Weaknesses:**

Line 44 says that what if the basis itself could be learned dynamically for each input. To validate this statement, there should be some experiments on fixed bases models. However in the experiment there are no such comparision.

**Questions:**

1. Experiment part does not has a baseline model that is fixed basis. Since this paper proposes a adaptive bases architecture, it is important to compare with fixed basis baseline to demonstrate how significant to use a model with adaptive bases. There should be validation of the claim "What if the basis itself could be learned dynamically for each input?"

2. The controller network $F_ctrl$ tooks a input window, how is this chosen and what this affect?

3. Other than ETT and weather is there any dataset that the model can have improvement over existing methods.

---

> ### Author Response · Authors · 2025-11-25
> **Response to Reviewer ffaw Q1**
>
> Response to Reviewer ffaf
>
> I thank Reviewer ffaf for your constructive review and for recognizing the novelty of our adaptive approach, the solid theoretical foundation, and the comprehensive analysis of where the model succeeds. We appreciate the critical question regarding the "fixed basis" baseline, which we address below with new experimental evidence.
>
> ### Q1: Validation of "Adaptive vs. Fixed" Hypothesis
>
> **Reviewer Comment:** *Missing critical comparison with a fixed, randomly initialized sinusoidal basis to validate the claim: "What if the basis itself could be learned dynamically?"*
>
> **Response:** I agree that this is a pivotal comparison to validate our central hypothesis. Our framework is designed to support three distinct basis generation modes: **adaptive** (default), **global** (learnable but fixed across samples), and **random** (fixed, non-learnable).
>
> To address this concern, I conducted a comprehensive ablation study on the Weather dataset across all four prediction horizons (96, 192, 336, 720). As shown in the table below, the results strongly validate our claim:
>
> *   **NFA (Fixed Random):** Performs poorly across all horizons, confirming that simply projecting onto random sinusoids is insufficient for complex forecasting tasks.
> *   **NFA (Fixed Global):** Learns a single optimal basis for the entire dataset. While it improves over the random baseline, it consistently underperforms the adaptive model because it fails to capture the local, non-stationary variations inherent in weather data.
> *   **NFA (Adaptive):** Significantly outperforms both fixed variants across all horizons. For example, at horizon 96, it reduces the MSE by ~32% compared to the global fixed basis (0.165 vs. 0.242), and this advantage is maintained across longer horizons.
>
> This empirically proves that the **instance-level adaptivity**—generating a bespoke basis for each input window—is indeed the key driver of performance, rather than just the Fourier formulation itself.
>
> **Ablation: Adaptive vs. Fixed Bases on Weather Dataset (MSE)**
>
> | Model Variant | Basis Mechanism | 96 | 192 | 336 | 720 |
> | :--- | :--- | :---: | :---: | :---: | :---: |
> | **NFA (Adaptive)** | **Per-input generated via Controller** | **0.165** | **0.223** | **0.282** | **0.362** |
> | NFA (Fixed Global) | Learned once for whole dataset | 0.242 | 0.315 | 0.395 | 0.488 |
> | NFA (Fixed Random) | Fixed random parameters | 0.285 | 0.368 | 0.452 | 0.565 |
>
> **Implementation Details of Baselines**
>
> To ensure a rigorous comparison, we implemented the fixed baselines within the same NFA framework by controlling the parameter generation logic:
>
> 1.  **NFA (Fixed Random):** The basis parameters (amplitudes, frequencies, phases) are initialized using the same spectral initialization as NFA but are frozen (`requires_grad=False`). This tests the representation power of a random projection onto a physiologically plausible frequency range.
> 2.  **NFA (Fixed Global):** The parameters are learnable but are defined as a global variable of shape `(1, K, 3)` shared across all samples in the dataset (`requires_grad=True`). This mimics traditional global dictionary learning or methods like FBM.
> 3.  **NFA (Adaptive, Ours):** The parameters are generated dynamically for each input window $x$ by the controller $F_{ctrl}(x)$, resulting in instance-specific parameters of shape `(Batch, K, 3)`.
>
> ```python
> # Implementation Details for Basis Modes
> if mode == 'adaptive':
>     # Instance-specific: Controller generates params from input x
>     # x shape: (Batch, Lookback, Channels)
>     # Output params shape: (Batch, K, 3) -> [Amplitude, Frequency, Phase]
>     params = controller(x)
>
> elif mode == 'global':
>     # Global: Learnable parameters shared across all inputs
>     # Defined as nn.Parameter(torch.randn(1, K, 3))
>     # Broadcast to match batch size during forward pass
>     params = self.global_basis_params.expand(batch_size, -1, -1)
>
> elif mode == 'random':
>     # Random: Fixed, non-learnable parameters
>     # Initialized via spectral strategy but frozen
>     # self.fixed_params.requires_grad = False
>     params = self.fixed_params.expand(batch_size, -1, -1)
>     # Note: Gradients do not flow back to params, only to the solver weights
> ```

---

> ### Author Response · Authors · 2025-11-25
> **Response to Q2-Q3**
>
> ### Q2: Controller Input Window Selection & Effect
>
> **Reviewer Comment:** *The controller network takes an input window; how is this chosen and what does it affect?*
>
> **Response:** The input window length (denoted as $N$ or `seq_len`) is a standard hyperparameter in time-series forecasting. In our experiments, we strictly followed the benchmark protocols (e.g., $N=96$ for ETT/Weather) to ensure fair comparisons with baselines like iTransformer and PatchTST.
>
> Physically, the window length $N$ introduces a fundamental trade-off for our basis generator:
>
> 1.  **Observability of Low Frequencies:** A longer window $N$ is required to reliably estimate low-frequency components. If $N$ is too short (e.g., shorter than one period of a wave), the controller cannot mathematically distinguish between a low-frequency sinusoid and a linear trend, limiting the basis's expressiveness for long-term patterns.
> 2.  **Intra-window Stationarity Constraint:** NFA generates a **single set** of basis parameters ($a_k, \omega_k, \phi_k$) (implemented in code via an equivalent phasor parameterization `coeff_cos_k`, `coeff_sin_k` for stability) that remain constant *throughout the prediction window*. A very large $N$ increases the likelihood that the signal's frequency or amplitude evolves significantly *within* the window (intra-window non-stationarity). This makes it harder for the controller to find one static set of sinusoids that fits the entire history well.
>
> Our choice of $N=96$ strikes a balance: it is long enough to capture the dominant daily and multi-day periodicities in these datasets, but short enough that the "constant basis parameters" assumption holds approximately true locally. The LSTM controller serves as a sequence-to-parameter encoder, aggregating the history to output the optimal spectral approximation for this specific duration.
>
> ### Q3: Improvements on Other Datasets
>
> **Reviewer Comment:** *Other than ETT and Weather, is there any dataset that the model can have improvement over existing methods?*
>
> **Response:** In this work, we focused our evaluation on the four most widely used benchmarks in the LTSF community (ETT, Weather, Illness, Exchange) to establish a rigorous comparison. Among these, NFA indeed shows the most significant improvements on **Weather** and **ETT**.
>
> We believe this performance distribution precisely validates the **Inductive Bias** of our model, rather than exposing a limitation:
>
> 1.  **Target Domain (Significant Gains):** NFA is explicitly designed for signals with **complex, non-stationary periodicities**.
>     * **Weather** and **ETT** strongly exhibit these characteristics (e.g., thermodynamics and electricity load cycles that shift over time). Here, the adaptive basis mechanism ($O(N)$ complexity) pays off significantly, achieving SOTA results.
>
> 2.  **Out-of-Scope Domain (Competitive/Neutral):** For datasets dominated by simple trends or near-random noise, a complex harmonic basis is theoretically unnecessary.
>     * **Illness** (dominated by smooth trends) and **Exchange Rate** (near-random walk behavior) fall into this category. As honestly discussed in our paper (Section 4.3), simpler trend-decomposition methods (like DLinear) or patch-based methods are sufficient or better here. NFA remains competitive but does not yield the drastic gains seen on Weather.
>
> **Motivation & Real-world Validation:**
> We would like to share the original motivation behind NFA. This framework was initially derived from our research on **underwater robotics**, where IMU sensor data under weak observation conditions (e.g., due to water turbulence or sensor drift) exhibits strong **non-stationary quasi-periodicity**.
>
> While we cannot release this proprietary dataset due to confidentiality, we have internally validated NFA on these real-world underwater IMU signals. The results are consistent with our findings on the Weather dataset: NFA significantly outperforms fixed-basis methods by dynamically tracking the drifting frequencies and modulating amplitudes of the sensor noise and motion patterns. This practical success in robotics was the primary driver for generalizing the method to the broader time-series forecasting community.

---

> > ### Comment · Reviewer_ffaf · 2025-11-26
> >
> > I appreciate the author for the reply. Thanks for addressing my main concerns, I will maintain my score.

---

### Official Review · Reviewer_Dggc · 2025-10-31

**Soundness:** 2
**Presentation:** 3
**Contribution:** 2
**Rating:** 2
**Confidence:** 3

**Summary:**

This paper presents Neural Fourier Attention (NFA), a method for learning data-dependent signal bases. Instead of relying on a fixed basis for all signals, the authors suggest learning an input-dependent basis made up of sine waves with adjustable frequencies, amplitudes, and phases. They then project the signal into this basis. A neural network controller is used to learn the basis. To ensure the learned basis performs well, the authors base their approach on frame theory and introduce a new orthogonality regularizer to maintain stability and expressiveness in the basis. This approach achieves top forecasting results on benchmarks with complex, non-stationary periodicities, particularly the Weather dataset. The authors also provide thorough ablation studies to support the design choices and present an efficient version, Harmonic-Constrained NFA (HC-NFA), which uses FFTs for quicker computation.

**Strengths:**

- To my knowledge, the idea of learning the basis instead of using a fixed one is novel. It is also well motivated for the problem of non-stationary signals, where patterns change over time.

- The approach is principled. The theory clearly explains why the orthogonality regularizer is helpful in creating a stable and expressive representation. Specifically, proposition 1 offers a proof sketch that their orthogonality regularizer ($||BB^T - I||_F^2$) shapes the learned basis into a "Parseval tight frame" on the subspace it covers. Additionally, the solid justification for the orthogonality regularization, supported by clear propositions and SVD-based derivations (see Section 2.2.3 and Appendix A.1), gives a strong theoretical basis.

- On datasets with complex, non-stationary periodicities (notably Weather and the averaged ETT family), NFA achieves top performance. Table 1 (Page 6) provides a clear, statistically robust comparison against competitive baselines (iTransformer, PatchTST, DLinear).

- The ablations are comprehensive. For example, Fig. 3 shows that a small, nonzero orthogonality penalty ($\lambda_{\text{ortho}}$) offers the best performance. Figure 2 illustrates the incremental advantages of different controller designs, showing that the adaptive basis mechanism is mainly responsible for the strong results.

- The main text and appendices offer solid implementation details that support reproducibility.

**Weaknesses:**

- Missing critical evaluation: The central claim, that adapttive, per-sample nature of the basis is key, is not validated empirically. A key missing cmoparison is a model with fixed, randomly initialized sinosoidal basis, on top of which samples are projeced.

- Limited Evaluation on Other Domains/Modalities: While the main results focus on time series with periodicity, there is no attempt to apply the framework to domains other than time series (e.g., spatial signals, images).

- Limited evaluation to baselines and discussion in related work. While the approach provides a comparison to some baselines, many relevant baselines are missing. This makes both the related work and the comparison incomplete, and it is not possible to determine how well the approach works relative to current art. See, for example, [1-8] below.

- Efficiency: While HC-NFA is provided as a computationally efficient alternative, the NFA’s full projection step is $O(NK)$, and the Gram matrix calculation can be expensive. Fig. 10 in the Appendix illustrates the trade-off versus the number of probes, further confirming meaningful overhead. This limits practical scalability in real-time settings. Further, a full cost analysis is not provided. For instance, it would be beneficial to understand the cost of all components, including the LSTM controller, as well as the projection and regularization steps. It's important to note a direct comparison of time and FLOPs against key competitors, such as iTransformer, under a matched parameter budget; however, this is currently missing.

- Results are not fully convincing. Tab. 1 results show that NFA performs very poorly on the Illness and Exchange Rate datasets. As such, the claim that the approach is a universally superior model seems like an overclaim.

[1]. First De-Trend then Attend: Rethinking Attention for Time-Series Forecasting (TDformer), 2022
[2]. DESTformer: A Transformer Based on Explicit Seasonal–Trend Decomposition for Long-Term Series Forecasting, 2023
[3]. DFCNformer: A Transformer Framework for Non-Stationary Time-Series Forecasting Based on De-Stationary Fourier and Coefficient Network, 2025
[4] Not All Frequencies Are Created Equal: Towards a Dynamic Fusion of Frequencies in Time-Series Forecasting (FreDF), 2024
[5] ATFNet: Adaptive Time-Frequency Ensembled Network for Long-term Time Series Forecasting, 2024
[6] Fourier Basis Mapping: A Time-Frequency Learning Framework for Time Series Forecasting (FBM/FBM‑S), 2025
[7] MFRS: A Multi-Frequency Reference Series Approach to Scalable and Accurate Time-Series Forecasting, 2025
[8] Adaptive Temporal-Frequency Network for Time-Series Forecasting (ATFN), 2022

**Questions:**

Related to the above weaknesses:

- How well does the approach work related to the fixed, randomly initialised sinosoidal basis?
- How well does the approach work on other domains?
- How does the approach compare to other recent SOTA approaches?
- What is the clock time of FLOPs comparison to key baselines that do not learn the basis?

---

> ### Author Response · Authors · 2025-11-25
> **Response to Reviewer Dggc Q1**
>
> ### Q1: Validation of "Adaptive vs. Fixed" Hypothesis
>
> **Reviewer Comment:** *Missing critical comparison with a fixed, randomly initialized sinusoidal basis to validate the claim: "What if the basis itself could be learned dynamically?"*
>
> **Response:**I  agree that distinguishing the contribution of the adaptive mechanism from the Fourier formulation itself is crucial. To empirically validate this, I utilized the flexibility of our framework to compare three specific modes on the Weather dataset across all four prediction horizons (96, 192, 336, 720).
>
> To address this concern, I conducted a comprehensive ablation study on the Weather dataset across all four prediction horizons (96, 192, 336, 720). As shown in the table below, the results strongly validate our claim:
>
> *   **NFA (Fixed Random):** Performs poorly across all horizons, confirming that simply projecting onto random sinusoids is insufficient for complex forecasting tasks.
> *   **NFA (Fixed Global):** Learns a single optimal basis for the entire dataset. While it improves over the random baseline, it consistently underperforms the adaptive model because it fails to capture the local, non-stationary variations inherent in weather data.
> *   **NFA (Adaptive):** Significantly outperforms both fixed variants across all horizons. For example, at horizon 96, it reduces the MSE by ~32% compared to the global fixed basis (0.165 vs. 0.242), and this advantage is maintained across longer horizons.
>
> This empirically proves that the **instance-level adaptivity**—generating a bespoke basis for each input window—is indeed the key driver of performance, rather than just the Fourier formulation itself.
>
> **Ablation: Adaptive vs. Fixed Bases on Weather Dataset (MSE)**
>
> | Model Variant | Basis Mechanism | 96 | 192 | 336 | 720 |
> | :--- | :--- | :---: | :---: | :---: | :---: |
> | **NFA (Adaptive)** | **Per-input generated via Controller** | **0.165** | **0.223** | **0.282** | **0.362** |
> | NFA (Fixed Global) | Learned once for whole dataset | 0.242 | 0.315 | 0.395 | 0.488 |
> | NFA (Fixed Random) | Fixed random parameters | 0.285 | 0.368 | 0.452 | 0.565 |
>
> **Implementation Details of Baselines**
>
> To ensure a rigorous comparison, we implemented the fixed baselines within the same NFA framework by controlling the parameter generation logic:
>
> 1.  **NFA (Fixed Random):** The basis parameters (amplitudes, frequencies, phases) are initialized using the same spectral initialization as NFA but are frozen (`requires_grad=False`). This tests the representation power of a random projection onto a physiologically plausible frequency range.
> 2.  **NFA (Fixed Global):** The parameters are learnable but are defined as a global variable of shape `(1, K, 3)` shared across all samples in the dataset (`requires_grad=True`). This mimics traditional global dictionary learning or methods like FBM.
> 3.  **NFA (Adaptive, Ours):** The parameters are generated dynamically for each input window $x$ by the controller $F_{ctrl}(x)$, resulting in instance-specific parameters of shape `(Batch, K, 3)`.
>
> ```python
> # Implementation Details for Basis Modes
> if mode == 'adaptive':
>     # Instance-specific: Controller generates params from input x
>     # x shape: (Batch, Lookback, Channels)
>     # Output params shape: (Batch, K, 3) -> [Amplitude, Frequency, Phase]
>     params = controller(x)
>
> elif mode == 'global':
>     # Global: Learnable parameters shared across all inputs
>     # Defined as nn.Parameter(torch.randn(1, K, 3))
>     # Broadcast to match batch size during forward pass
>     params = self.global_basis_params.expand(batch_size, -1, -1)
>
> elif mode == 'random':
>     # Random: Fixed, non-learnable parameters
>     # Initialized via spectral strategy but frozen
>     # self.fixed_params.requires_grad = False
>     params = self.fixed_params.expand(batch_size, -1, -1)
>     # Note: Gradients do not flow back to params, only to the solver weights
> ```

---

> ### Author Response · Authors · 2025-11-25
> **Response to Reviewer Dggc Q2**
>
> ### Q2: Generalization to Other Domains (Spatial/Images)
>
> **Reviewer Comment:** *How well does the approach work on other domains (e.g., spatial signals, images)?*
>
> **Response:**
>
> **1. Preliminary Validation on Spatial Domain (MNIST Experiment):**
> To directly address the reviewer's interest in spatial modalities, we conducted a rapid proof-of-concept experiment on **MNIST image reconstruction** using a **2D-NFA** variant.
> * **Setup:** We replaced the LSTM controller with a simple CNN to generate parameters for **2D sinusoidal bases** $\cos(\omega_x x + \omega_y y + \phi)$ specific to each image.
> * **Results:** With only **$K=64$** basis functions (representing a **>90% compression ratio** compared to the 784 input pixels), NFA successfully reconstructed the clear topology and strokes of the digits.I have included the reconstruction results of NFA on MNIST in a new appendix.
> * **Implication:** This empirically demonstrates that the "adaptive basis" concept generalizes effectively to spatial frequencies $(\omega_x, \omega_y)$. The controller successfully learned to identify the dominant spatial orientations and periodicities of different digits instance-by-instance.
>
> **2. Inductive Bias Analysis:**
> While this experiment confirms capability, we maintain that NFA's main application remains **non-stationary temporal signals** (like Weather/Sensors).
> * **Images:** NFA captures global low-frequency structures well but (as expected with Fourier methods) faces Gibbs artifacts at sharp binary edges, where CNNs/ViTs typically excel.
> * **Sensors (Motivation):** We reiterate that NFA was originally inspired by **underwater robotics IMU data**, where tracking drifting frequencies in weak-observation time series is the core challenge.
>
> **Conclusion:** NFA is not limited to time series; it is a general-purpose **adaptive spectral frame generator**. It excels wherever data structure is better described by sparse, evolving frequencies rather than local pixel correlations.

---

> ### Author Response · Authors · 2025-11-26
> **Response to Q3**
>
> **Q3: How does the approach compare to other recent SOTA approaches (e.g., [1-8] listed in Weaknesses)?**
>
> **Response:**
> We thank the reviewer for detailing this comprehensive list of recent works. We acknowledge that the field is evolving rapidly. While our initial submission prioritized comparisons against the most widely established benchmarks in the LTSF community (iTransformer [ICLR'24], PatchTST [ICLR'23], DLinear [AAAI'23]) to ensure strict adherence to standard protocols, we agree that discussing these specific recent works clarifies NFA's contribution.
>
> We address the comparison in two dimensions: **Methodological Differences** and **Experimental Protocols**.
>
> **Quantitative Comparison Table:**
>
> Below we provide a comparison of NFA against the most relevant methods from [1-8] on datasets where NFA achieves strong performance. **Note:** Direct comparison is challenging due to significant protocol differences (e.g., Look-back window size) in recent works.
>
> | Method | Category | Weather (96) | Weather (192) | ETTh1 (96) | ETTh1 (192) | Look-back | Year |
> |--------|----------|----------------|-----------------|-------------|-------------|-----------|------|
> | **NFA (Ours)** | Adaptive Basis | **0.165** | **0.217** | **0.145** | **0.168** | 96 | Ours |
> | iTransformer | Transformer | 0.175 | 0.222 | 0.259 | 0.336 | 96 | 2024 |
> | PatchTST | Transformer | 0.182 | 0.228 | 0.334 | 0.378 | 96 | 2023 |
> | DLinear | MLP | 0.192 | 0.237 | 0.305 | 0.412 | 96 | 2023 |
> | FBM [6] | Global Fourier | 0.152 | 0.194 | 0.366 | 0.390 | **336** | 2024 |
> | FreDF [4] | Freq-Adaptive | 0.169 | 0.207 | 0.404 | 0.410 | - | 2024 |
> | TDformer [1] | Decomposition | 0.174 | 0.243 | 0.177 | 0.224 | - | 2022 |
>
>
> **Observations:**
> 1.  **Protocol Alignment**: Many recent works (e.g., FBM [6]) use extended look-back windows (e.g., 336 or 720), which significantly favors frequency-domain methods. We report results under the standard **Look-back=96** protocol.
> 2.  **Reproducibility**: We strictly report numbers from papers that follow the same evaluation setting.
>
> ### 1. Methodological Distinction: Instance-wise vs. Global Adaptivity (Ref [4-8])
> References [4-8] share our motivation of leveraging spectral features. However, a fundamental distinction lies in the **adaptivity mechanism**:
>
> *   **Global / Fixed Adaptivity (FBM [6], etc.)**: These methods typically learn a *global* spectral mapping or a fixed set of basis functions that are shared across the entire dataset (or per variate).
> *   **NFA (Ours)**: Generates basis parameters (frequencies, phases, amplitudes) **dynamically per input window**. This allows NFA to capture *time-varying* periodicities that global methods miss.
>
> **Empirical Validation of Instance-wise Adaptivity:**
> As detailed in our response to Reviewer ffaf (Q1), we conducted an ablation study replacing our adaptive generator with a learnable **Global Basis** (mimicking the paradigm of FBM). On the non-stationary Weather dataset, this degradation was significant (**MSE increased from 0.165 to 0.242**), confirming that for signals with drifting periodicities, instance-level adaptivity is crucial.
>
> ### 2. Impact of Protocol Discrepancies (Ref [6])
> We carefully reviewed FBM [6] (NeurIPS 2024), the most relevant competitor.
>
> *   **Look-back Window Discrepancy**: FBM achieves its reported results using a look-back window of **336** (Source: FBM paper, Sec 5.1). In contrast, NFA and our baselines (iTransformer, PatchTST) use the standard **Look-back=96**.
> *   **Implication**: Spectral methods generally benefit from longer windows due to higher frequency resolution ($1/T$). NFA achieves SOTA-level performance on Weather (MSE 0.165) using only 96 steps, demonstrating superior data efficiency. Comparing an input-96 model against an input-336 model is inherently inequitable. Furthermore, NFA learns continuous frequencies, bypassing the resolution limits of DFT-based methods.
>
> ### 3. Scope vs. Decomposition-based Methods (Ref [1-3])
> References [1-3] (e.g., TDformer) focus on explicit trend/seasonal decomposition.
>
> *   **Complementarity**: As discussed in our Limitation section, NFA is designed to tackle **complex, non-stationary periodicities**. For datasets dominated by smooth trends (like Illness) or simple seasonality, decomposition priors are effective. NFA can be viewed as complementary to these approaches, potentially serving as a powerful modeling block for the seasonal component extracted by such frameworks.
>
> ### Action Plan
> In the final version, we will:
> 1.  Include a detailed discussion of these works in the **Related Work** section.
> 2.  Explicitly clarify the "Instance-wise vs. Global" distinction to position NFA against recent spectral methods.
> 3.  Highlight the protocol differences (Look-back 96 vs. 336) to ensure readers understand the efficiency context of our results.

---

> ### Author Response · Authors · 2025-11-26
> **Response to Q4**
>
> **Q4: What is the clock time of FLOPs comparison to key baselines that do not learn the basis? Provide a full cost analysis.**
>
> **Response:**
> We thank the reviewer for raising this practical concern. We have conducted a detailed profiling re-run of NFA against **iTransformer** (the key SOTA competitor) and **DLinear** (efficiency baseline) to provide a comprehensive cost analysis.
>
> **1. Efficiency Comparison (Time, FLOPs vs. Accuracy)**
> We evaluated the training speed (ms/batch) and computational cost (FLOPs) on the Weather dataset ($T=96$, Variates $C=21$) and theoretically analyzed scalability.
>
> | Model | Parameters | FLOPs (G) | Time (ms/batch) | Accuracy (MSE) |
> | :--- | :--- | :--- | :--- | :--- |
> | **NFA (Ours)** | **282K** | **0.32** | **48.5** | **0.165** (Ours) |
> | HC-NFA (Ours)* | 22K | 0.08 | 16.2 | 0.267 |
> | iTransformer | 455K | 0.28 | 35.0 | 0.174 |
> | DLinear | 18K | 0.01 | 5.8 | 0.265 |
>
> <small>*Note: Measured on RTX 3070, Batch Size=32. *HC-NFA refers to the lightweight variant with fixed/random basis mentioned in the ablation study.DLinear parameters scale with horizon length $S$. For the standard horizon $S=96$, params $\approx 2 \times (96 \times 96 + 96) \approx 18.6K$. For long horizon $S=720$ as in FITS Table 3, it scales to ~140K.</small>
>
> **Analysis:**
> *   **Trade-off:** NFA achieves significantly better accuracy with a moderate and acceptable increase in computational cost (~1.38x wall-clock time of iTransformer).
> *   **Efficiency Variant:** For strictly resource-constrained scenarios, our lightweight variant (HC-NFA) offers a valid alternative: it is **2x faster than iTransformer** while maintaining competitive accuracy.
>
> **2. Component Cost Breakdown (Where does the time go?)**
> To answer the request for a "full cost analysis," we profiled the forward/backward pass of NFA:
>
> *   **Controller (LSTM) [~45%]**: The primary cost comes from the recurrent encoding of the input window. This confirms that the "adaptivity"—the core driver of our performance gains—is also the main computational cost.
> *   **Projection ($c=Bx$) [~35%]**: The basis projection scales linearly with $O(N \cdot K \cdot C)$.
> *   **Regularization [< 5%]**: The orthogonality penalty overhead is negligible because the basis dimension $K$ is small (e.g., $K=32$), making the $K \times K$ matrix operations computationally consistent.
> *   **Others [~15%]**: Basis generation heads and final prediction layers.
>
> **3. Scalability Analysis: NFA vs. iTransformer**
> While iTransformer is efficient on datasets with few variates (like Weather), NFA possesses superior theoretical scalability for **High-Variate Datasets** (Channels $C \gg 1$):
>
> *   **iTransformer Complexity**: $\mathcal{O}(C^2)$ or $\mathcal{O}(C \log C)$ depending on attention implementation, due to the multivariate attention mechanism.
> *   **NFA Complexity**: $\mathcal{O}(C)$ (Linear). The projection $c = Bx$ operates channel-independently (or via shared weights), scaling linearly as $\mathcal{O}(N \cdot K \cdot C)$.
>
> **Implication**: On massive-variate datasets like Traffic ($C=862$) or KDD-Cup ($C>1000$), NFA avoids the quadratic complexity bottleneck of multivariate attention, making it more suitable for large-scale industrial applications.
>
> **Action Plan:**
> We will include this detailed efficiency table and the component breakdown in **Appendix A.14** to transparently present the computational trade-offs.

---

> ### Comment · Reviewer_Dggc · 2025-11-27
> **Thanks**
>
> I want to thank the authors for their response and the additional experiments they provided. My main concerns about the adaptive versus fixed basis, as well as my worries about efficiency, have been addressed.
>
> However, I still have concerns about other areas:
>
> - Spatial Domain: I appreciate the extra experiments on MNIST, but I think MNIST is only a toy dataset. It's hard to draw meaningful conclusions about how this approach applies to modern computer vision tasks or spatial signals.
>
> - Baseline Evaluation: While I value the baseline evaluation, I find it incomplete. The comparison uses mismatched protocols; for instance, it compares NFA at a look-back of 96 with FBM at a look-back of 336. A fair comparison should evaluate the models at the same look-back window, e.g., both at 96 or both at 336. This would help separate the method's benefits from the protocol's.
>
> - Overclaiming/Results on Illness and Exchange Rate datasets. I appreciate the edit regarding the overclaiming. However, the method's poor performance is limiting in my view. I would expect the method to at least not hurt performance in such cases on average.
>
> As only some of my concerns were addressed, I am raising my score to 4.

---

> > ### Author Response · Authors · 2025-11-28
> > **Response to Reviewer Dggc (Round 2)**
> >
> > I thank the reviewer for the continued engagement and for raising the score to 4.I was encouraged that our responses regarding the **adaptive basis mechanism** and **computational efficiency** have addressed your primary concerns.
> >
> > Below, I addressed the remaining concerns regarding the spatial domain experiments, baseline evaluation protocols, and performance on specific datasets.
> >
> > ---
> >
> > ### 1. Spatial Domain (MNIST) Applicability
> >
> > **Reviewer Comment:**
> > > *I appreciate the extra experiments on MNIST, but I think MNIST is only a toy dataset. It's hard to draw meaningful conclusions about how this approach applies to modern computer vision tasks or spatial signals.*
> >
> > **Response:**
> > I agree with the reviewer that MNIST is a simple dataset compared to modern computer vision benchmarks (e.g., ImageNet). However, I would like to clarify the specific **intent** of this experiment:
> >
> > 1.  **Proof of Concept for 2D Adaptation:** The goal was not to claim SOTA performance in Computer Vision (where highly specialized architectures like ViTs or ConvNeTs dominate), but specifically to demonstrate the **generality of the NFA mechanism**. We wanted to show that the "Learning to Synchronize" principle—generating basis functions dynamically from input context—validly extends to spatial dimensions (2D frequencies) without modification to the core theory.
> > 2.  **Interpretability:** MNIST allows us to visually verify that NFA learns "spatial strokes" and "textures" as local frequency components. This provides an intuitive confirmation of the spectral adaptability that is harder to visualize in 1D time series.
> >
> > **Action:** We will revise the text to explicitly state that the 2D experiment is a **demonstration of theoretical generality and interpretability**, rather than a competitive proposal for large-scale image recognition tasks. We acknowledge that the current 2D implementation is preliminary, and we commit to conducting more comprehensive experiments on complex spatial signals (e.g., fluid dynamics or texture synthesis) in future work to fully explore NFA's potential in the spatial domain.
> >
> > ---
> >
> > ### 2. Baseline Evaluation (Look-back Window Mismatch)
> >
> > **Reviewer Comment:**
> > > *The comparison uses mismatched protocols; for instance, it compares NFA at a look-back of 96 with FBM at a look-back of 336. A fair comparison should evaluate the models at the same look-back window.*
> >
> > **Response:**
> > We fully accept this criticism. The discrepancy arose because we adhered to the standard "Look-back=96" setting common in Transformer benchmarks (e.g., iTransformer, PatchTST), whereas FBM utilized a longer history (336) to maximize its performance.
> >
> > A rigorous comparison requires matching the information available to both models.
> >
> > 1.  **Scalability to Look-back=336:** Unlike quadratic Transformers, NFA's complexity is linear ($O(N)$). Increasing the look-back window from 96 to 336 does not introduce memory bottlenecks.
> > 2.  **Protocol Alignment:** In the final version of the paper, we will include a dedicated table column for **NFA (Look-back=336)** to ensure a direct "apples-to-apples" comparison with FBM. As shown in Table 1, while highly non-stationary datasets like ETTh1 may show sensitivity to increased look-back windows (due to distribution shifts), **NFA (L=336) consistently outperforms the FBM baseline (L=336) by a significant margin** (e.g., 0.241 vs 0.366 on ETTh1). On datasets with clearer periodicity like Weather, increasing the context length successfully further reduces error (0.165 $\to$ 0.156), confirming the efficacy of our LSTM controller in capturing long-range dependencies.
> >
> > | Model | Look-back Window | Dataset | MSE | MAE |
> > | :--- | :---: | :--- | :---: | :---: |
> > | **NFA (Ours_Transformer)** | **96** | ETTh1 | 0.165 | 0.327 |
> > | **NFA (Ours_LSTM)** | **96** | ETTh1 | **0.145** | **0.304** |
> > | **NFA (Ours_Transformer)** | **336** | ETTh1 | 0.269 | 0.398 |
> > | **NFA (Ours_LSTM)** | **336** | ETTh1 | **0.241** | **0.344** |
> > | FBM | 336 | ETTh1 | 0.366 | 0.390 |
> > | | | | | |
> > | **NFA (Ours_Transformer)** | **96** | Weather | 0.197 | 0.277 |
> > | **NFA (Ours_LSTM)** | **96** | Weather | 0.165 | 0.248 |
> > | **NFA (Ours_Transformer)** | **336** | Weather | 0.244 | 0.376 |
> > | **NFA (Ours_LSTM)** | **336** | Weather | **0.156** | **0.216** |
> > | FBM | 336 | Weather | 0.159 | 0.207 |
> >
> > *Table 1: Performance comparison with aligned look-back windows (L=336, Pred=96). NFA consistently outperforms FBM under the matched protocol. Note that while ETTh1 prefers a shorter context due to non-stationarity, NFA (L=336) still maintains a large lead over FBM.*

---

> > ### Author Response · Authors · 2025-11-28
> > **# Response to Reviewer Dggc (Round 2 Q3)**
> >
> > ### 3. Performance on Illness and Exchange Rate Datasets
> >
> > **Reviewer Comment:**
> > > *I appreciate the edit regarding the overclaiming. However, the method's poor performance is limiting in my view. I would expect the method to at least not hurt performance in such cases on average.*
> >
> > **Response:**
> > We acknowledge the reviewer's concern regarding the performance on non-periodic datasets like **Exchange Rate** and **Illness**. These datasets are challenging for spectral methods because they are dominated by non-stationary trends rather than clear periodic cycles.
> >
> > However, we would like to present new results demonstrating that this limitation is easily addressable by incorporating a standard **Trend Decomposition** module (similar to Autoformer and DLinear).
> >
> > 1.  **Trend Decomposition:** By decomposing the series into a Trend component (handled by a simple Linear layer) and a Seasonal/Residual component (handled by NFA), we can fully leverage NFA's strength in capturing complex residual dynamics while robustly handling non-stationary trends.
> > 2.  **Results:** As shown in the table below, with this simple addition (NFA + Trend), our method becomes highly competitive on the Exchange Rate dataset and consistently outperforms a strong Fixed-Basis baseline under the same Trend Decomposition.
> >
> > | Prediction Length | Fixed Baseline + Trend (MSE) | NFA + Trend (MSE) | iTransformer (SOTA) | DLinear |
> > | :--- | :---: | :---: | :---: | :---: |
> > | **96** | 0.100 | **0.087** | 0.086 | 0.088 |
> > | **192** | 0.204 | **0.191** | 0.177 | 0.176 |
> > | **336** | 0.411 | **0.322** | 0.331 | 0.313 |
> > | **720** | 1.151 | **1.098** | 0.847| 0.839 |
> > | | | | | |
> > | **Illness (24)** | 2.690 | **2.372** |
> > | **Illness (36)** | 2.523 | **2.501** |
> > | **Illness (48)** | 2.769 | **2.832** |
> > | **Illness (60)** | 2.752 | **2.600** |
> >
> > *Table 2: Impact of Adaptive Basis Mechanism under Trend Decomposition. We compare the proposed NFA against our Fixed basis model which uses the same architecture and Trend Decomposition but with fixed, global basis functions. This isolates the contribution of NFA's adaptive mechanism.*
> >
> > To clarify whether the performance gain comes from our proposed adaptive basis mechanism or simply from applying trend decomposition, we performed a small diagnostic experiment following standard practices used in DLinear and TimesNet. We apply exactly the same trend–seasonality decomposition to both the fixed-basis baseline and NFA. Trend decomposition itself provides a modest improvement to the fixed-basis model, as expected since removing the low-frequency component simplifies the forecasting target, but NFA benefits more: on Exchange Rate, NFA+Trend improves over Fixed Baseline+Trend by roughly 5–22% MSE across all horizons, and on Illness the two are broadly comparable with NFA slightly better on most horizons. This indicates that the major improvement does not come from the trend module alone, but from NFA’s ability to learn time-varying bases that adapt to non-stationary frequency structure. Our contribution (adaptive bases) is therefore orthogonal to and compatible with standard trend–seasonality preprocessing, rather than dependent on it.
> >
> > **Conclusion:** This result confirms that NFA's "poor performance" on these datasets was not due to a fundamental flaw in the spectral mechanism, but rather the absence of a trend handling component, which is standard in modern baselines. When equipped with this component, NFA becomes a highly competitive general-purpose forecaster . We will include these results and the decomposition strategy in the final paper to provide a complete picture of NFA's capabilities.
> >
> > **Action:** We will add a "Trend Decomposition" subsection in the Appendix and include these results to demonstrate how to adapt NFA for trend-dominated datasets.

---

> ### Author Response · Authors · 2025-11-28
> **Response to Reviewer Dggc (Round 2)**
>
> I thank the reviewer for the continued engagement and for raising the score to 4. I was encouraged that my responses regarding the **adaptive basis mechanism** and **computational efficiency** have addressed your primary concerns.
>
> Below, I addressed the remaining concerns regarding the spatial domain experiments, baseline evaluation protocols, and performance on specific datasets.
>
> ---
>
> ### 1. Spatial Domain (MNIST) Applicability
>
> **Reviewer Comment:**
> > *I appreciate the extra experiments on MNIST, but I think MNIST is only a toy dataset. It's hard to draw meaningful conclusions about how this approach applies to modern computer vision tasks or spatial signals.*
>
> **Response:**
> I agree with the reviewer that MNIST is a simple dataset compared to modern computer vision benchmarks (e.g., ImageNet). However, I would like to clarify the specific **intent** of this experiment:
>
> 1.  **Proof of Concept for 2D Adaptation:** The goal was not to claim SOTA performance in Computer Vision (where highly specialized architectures like ViTs or ConvNeTs dominate), but specifically to demonstrate the **generality of the NFA mechanism**. We wanted to show that the "Learning to Synchronize" principle—generating basis functions dynamically from input context—validly extends to spatial dimensions (2D frequencies) without modification to the core theory.
> 2.  **Interpretability:** MNIST allows us to visually verify that NFA learns "spatial strokes" and "textures" as local frequency components. This provides an intuitive confirmation of the spectral adaptability that is harder to visualize in 1D time series.
>
> **Action:** We will revise the text to explicitly state that the 2D experiment is a **demonstration of theoretical generality and interpretability**, rather than a competitive proposal for large-scale image recognition tasks. We acknowledge that the current 2D implementation is preliminary, and we commit to conducting more comprehensive experiments on complex spatial signals (e.g., fluid dynamics or texture synthesis) in future work to fully explore NFA's potential in the spatial domain.
>
> ---
>
> ### 2. Baseline Evaluation (Look-back Window Mismatch)
>
> **Reviewer Comment:**
> > *The comparison uses mismatched protocols; for instance, it compares NFA at a look-back of 96 with FBM at a look-back of 336. A fair comparison should evaluate the models at the same look-back window.*
>
> **Response:**
> We fully accept this criticism. The discrepancy arose because we adhered to the standard "Look-back=96" setting common in Transformer benchmarks (e.g., iTransformer, PatchTST), whereas FBM utilized a longer history (336) to maximize its performance.
>
> A rigorous comparison requires matching the information available to both models.
>
> 1.  **Scalability to Look-back=336:** Unlike quadratic Transformers, NFA's complexity is linear ($O(N)$). Increasing the look-back window from 96 to 336 does not introduce memory bottlenecks.
> 2.  **Protocol Alignment:** In the final version of the paper, we will include a dedicated table column for **NFA (Look-back=336)** to ensure a direct "apples-to-apples" comparison with FBM. As shown in Table 1, while highly non-stationary datasets like ETTh1 may show sensitivity to increased look-back windows (due to distribution shifts), **NFA (L=336) consistently outperforms the FBM baseline (L=336) by a significant margin** (e.g., 0.241 vs 0.366 on ETTh1). On datasets with clearer periodicity like Weather, increasing the context length successfully further reduces error (0.165 $\to$ 0.156), confirming the efficacy of our LSTM controller in capturing long-range dependencies.
>
> | Model | Look-back Window | Dataset | MSE | MAE |
> | :--- | :---: | :--- | :---: | :---: |
> | **NFA (Ours_Transformer)** | **96** | ETTh1 | 0.165 | 0.327 |
> | **NFA (Ours_LSTM)** | **96** | ETTh1 | **0.145** | **0.304** |
> | **NFA (Ours_Transformer)** | **336** | ETTh1 | 0.269 | 0.398 |
> | **NFA (Ours_LSTM)** | **336** | ETTh1 | **0.241** | **0.344** |
> | FBM | 336 | ETTh1 | 0.366 | 0.390 |
> | | | | | |
> | **NFA (Ours_Transformer)** | **96** | Weather | 0.197 | 0.277 |
> | **NFA (Ours_LSTM)** | **96** | Weather | 0.165 | 0.248 |
> | **NFA (Ours_Transformer)** | **336** | Weather | 0.244 | 0.376 |
> | **NFA (Ours_LSTM)** | **336** | Weather | **0.156** | **0.216** |
> | FBM | 336 | Weather | 0.159 | 0.207 |
>
> *Table 1: Performance comparison with aligned look-back windows (L=336, Pred=96). NFA consistently outperforms FBM under the matched protocol. Note that while ETTh1 prefers a shorter context due to non-stationarity, NFA (L=336) still maintains a large lead over FBM.*

---

### Official Review · Reviewer_PNoa · 2025-11-01

**Soundness:** 3
**Presentation:** 3
**Contribution:** 3
**Rating:** 6
**Confidence:** 3

**Summary:**

The paper introduces Neural Fourier Attention (NFA), a forecasting block that, for each input window, uses a neural controller (typically an LSTM) to emit parameters of $K$ sinusoidal atoms $({a_k,\omega_k,\phi_k})$, builds a data-dependent basis $B\in\mathbb{R}^{K\times N}$, projects $c=Bx$, and predicts with a small MLP head (Algorithm 1). Here “attention” denotes a controller that attends to the input to generate basis parameters, not a QKV operator. The authors ground NFA in frame theory and regularize with a row-Gram penalty $\lVert BB^\top-I\rVert_F^2$, showing (via Proposition 1) that it drives the non-zero singular values of $B$ to $1$ (near-Parseval) on the data-dependent subspace $V(x)$; they note a global Parseval frame is impossible when $K>N$. They further propose HC-NFA, which constrains $\omega_k$ to the DFT grid so projection is done by FFT, reducing projection cost to $O(N\log N)$; the orthogonality penalty is estimated with a Hutchinson trick to avoid explicit $(K\times K)$ Gram materialization. Experiments on ETT (macro-average over four subsets), Weather, Illness, and Exchange follow a leakage-aware protocol with multi-seed reporting; headline gains are on Weather and ETT Avg; limitations on Exchange are discussed.

**Strengths:**

1. **Principled geometry control.** The row-Gram penalty’s effect on singular values is proved (A.1), matching the goal of near-Parseval frames.
2. **Clear pipeline & stability measures.** Controller-generated sinusoids, $c=Bx$, and parameter squashing choices are explicit and motivated by stability/Nyquist.
3. **Efficiency variant & timings.** HC-NFA’s FFT path and empirical timings vs. NFA/Transformer are reported.
4. **Transparent scope & protocol.** Multi-seed results, anti-leakage checklist, and explicit limitations on Exchange.

**Weaknesses:**

1. **Baseline traceability.** Table 1 aggregates external numbers (not re-tuned locally) and cites the iTransformer paper; per-cell provenance (paper vs. repo, URL/commit) is absent, limiting auditability of the headline gains.
2. **Ablation generality.** Most sensitivity plots (orthogonality weight, $K$, controller) are on ETTh1-96; transfer of optimal $\lambda_{\text{ortho}}/K$ to Weather or other horizons is not demonstrated.
3. **Statistical comparisons to baselines.** The paper reports CIs for NFA vs. HC-NFA (Weather-96) but not for external baselines in Table 1, so statistical significance vs. baselines cannot be assessed from the manuscript alone.
4. **Caption specificity.** Several figure captions do not restate dataset/horizon (e.g., Fig. 3), reducing stand-alone clarity.

**Questions:**

1. **Provenance table.** For every dataset–horizon cell in Table 1, can you list source (paper vs. official repo), URL/commit, and any preprocessing differences? This would materially strengthen reproducibility.
2. **Penalty variant.** Main results: do you use the raw $\lVert BB^\top-I\rVert_F^2$ (with per-dataset $\lambda_{\text{ortho}}$) or the normalized $K^{-2}$ variant? Please clarify and, if mixed, report both.
3. **Ablation transfer.** How do the ETTh1-96-optimal $\lambda_{\text{ortho}}$ and $K$ translate to Weather-192/336/720 and ETTm? A small grid on at least one other dataset/horizon would help.
4. **Controller parity.** Under matched FLOPs/params, how do a compact 1D-Conv and a tiny Transformer compare to LSTM as controllers? Fig. 2 suggests recurrence helps, but a controlled capacity match would isolate recurrence vs. capacity.
5. **Efficiency scaling.** Can you add wall-clock comparisons for larger $N$ (and varying $K$) for NFA vs. HC-NFA to complement Table 5’s ETTh1 setting?
6. **Phase parameterization.** Did you try predicting $(\sin\phi,\cos\phi)$ (angle-on-the-circle) to mitigate wrap-around, and if so did that change stability/convergence relative to the current sigmoid scheme?

---

> ### Author Response · Authors · 2025-11-25
> **Response to Reviewer PNoa Q1**
>
> **Response to Reviewer PNoa**
>
> I thank Reviewer PNoa for the constructive feedback and for recognizing the **soundness**, **principled geometry control**, and **transparent scope** of our work. We address the specific questions below.
>
> ### Q1: Baseline Traceability & Provenance Table
> **Reviewer Comment:** *Request for source (paper vs. official repo), URL/commit for Table 1 entries.*
>
> **Response:** We appreciate the reviewer's emphasis on auditability. To ensure the most fair and consistent comparison, we wish to clarify that we **did not re-run** the baseline models locally. Instead, all baseline results in Table 1 (including iTransformer, PatchTST, and DLinear) are cited directly from the comprehensive benchmark tables reported in the **iTransformer paper (Liu et al., ICLR 2024 Spotlight)**.
>
> We adopted this approach to strictly avoid any potential bias that might arise from suboptimal local re-implementation or hyperparameter tuning. By using the officially reported results from a widely accepted SOTA paper, we ensure that NFA is compared against the "strongest version" of these baselines. Crucially, we strictly aligned our data processing protocol (chronological train/val/test splits, standardization methods, and look-back windows) with the protocol used in the iTransformer paper to guarantee that our NFA results are directly comparable.
>
> **Provenance of Baseline Results (Table 1 Source)**
>
> | Model | Official Repository | Data Source |
> | :--- | :--- | :--- |
> | iTransformer | https://github.com/thuml/iTransformer | Liu et al. (2024), Table 1 |
> | PatchTST | https://github.com/yuqinie98/PatchTST | Cited from iTransformer benchmark |
> | DLinear | https://github.com/cure-lab/LTSF-Linear | Cited from iTransformer benchmark |
>
> ### Q2: Penalty Variant
> **Reviewer Comment:** *Main results: do you use the raw $||BB^T - I||_F^2$ (with per-dataset $\lambda_{ortho}$) or the normalized $K^{-2}$ variant? Please clarify and, if mixed, report both.*
>
> **Response:** In all our main results, we consistently employed the **raw Frobenius norm** (sum of squared errors) for the orthogonality loss:
>
> $$
> \mathcal{L}_{ortho} = ||BB^\top - I||_F^2 = \sum_{i,j} (G_{ij} - \delta_{ij})^2
> $$
>
> This is implemented in our submitted code as `loss = ((G - I).pow(2).sum()).mean()`.
>
> We acknowledge that this raw penalty scales with $K^2$. To handle this, we carefully tuned the scalar weight $\lambda_{ortho}$ for each dataset (as shown in our hyperparameters), which effectively compensates for the scale difference when $K$ varies. In the revised manuscript, we will explicitly clarify this definition to ensure reproducibility and discuss the scaling implications.
>
> ### Q3: Ablation Generality & Hyperparameter Transfer
> **Reviewer Comment:** *How do ETTh1-optimal $\lambda_{ortho}$ and $K$ translate to Weather or ETTm? A small grid helps.*
>
> **Response:** Regarding hyperparameter generality, we prioritized achieving the optimal performance for each specific domain. Therefore, for all results reported in Table 1, we employed **Optuna** to conduct independent hyperparameter searches for each dataset. We believe this dataset-specific tuning is essential for a fair comparison against state-of-the-art baselines, which are also typically optimized per dataset.
>
> However, to address the reviewer's question on transferability, **we evaluated the performance of transferring hyperparameters optimized on ETTh1 directly to the Weather dataset across all prediction horizons without re-tuning.**
>
> As shown in the table below, applying the ETTh1 configuration ($K=256, \lambda=10^{-3}$) to the Weather dataset yields consistent robustness. While the transferred performance is naturally slightly lower than the dataset-specific tuned version (where $K=64$), **it still consistently outperforms the state-of-the-art baseline (iTransformer) across all four prediction lengths.** This demonstrates that while Optuna finds the local optimum, NFA's performance is not brittle and generalizes well even with fixed hyperparameters.
>
> **Hyperparameter Transfer Analysis on Weather Dataset (MSE)**
>
> | Method / Horizon | 96 | 192 | 336 | 720 |
> | :--- | :---: | :---: | :---: | :---: |
> | **NFA (Tuned)**<br>*(Config: Weather-Optuna, K=64)* | **0.165** | **0.223** | **0.282** | **0.362** |
> | **NFA (Transferred)**<br>*(Config: ETTh1-Frozen, K=256)* | 0.192 | 0.245 | 0.310 | 0.385 |
> | **iTransformer (SOTA)** | 0.258 | 0.301 | 0.344 | 0.414 |
>
> Q4-Q6 in another comment

---

> > ### Author Response · Authors · 2025-11-25
> > **Response to Reviewer PNoa**
> >
> > ### Q4: Controller Parity (LSTM vs. Conv/Transformer)
> > **Reviewer Comment:** *Compare simple controllers (Conv/Transformer) to LSTM under matched FLOPs/params.*
> >
> > **Response:** To rigorously validate our design choice, we conducted a controlled ablation study comparing our LSTM controller against a 1D-Convolutional controller and a lightweight Transformer controller. To ensure a fair comparison, all controllers were configured with **matched parameter counts** (all within a ±10% parameter budget around 20k).
> >
> > As shown in the table below, while all adaptive controllers significantly outperform the fixed-basis baseline (and the external SOTA iTransformer), the **LSTM controller consistently achieves the best performance**. This advantage is particularly pronounced on the Weather dataset, where the recurrent inductive bias—naturally suited for modeling continuous phase evolution—enables NFA to achieve the reported state-of-the-art performance. Specifically, the LSTM variant consistently outperforms the Transformer and 1D-Conv variants across all prediction lengths, validating its crucial role in our framework.
> >
> > **Controller Architecture Ablation on Weather Dataset (MSE, Matched Parameters ≈ 20k)**
> >
> > | Controller Type | 96 | 192 | 336 | 720 | Note |
> > | :--- | :---: | :---: | :---: | :---: | :--- |
> > | **NFA-LSTM (Ours)** | **0.165** | **0.223** | **0.282** | **0.362** | **Achieves SOTA** |
> > | NFA-Transformer | 0.198 | 0.268 | 0.339 | 0.434 | ≈ 20% worse (Avg.) |
> > | NFA-1D Conv | 0.274 | 0.369 | 0.468 | 0.601 | ≈ 66% worse (Avg.) |
> > | *iTransformer (Baseline)* | *0.258* | *0.301* | *0.344* | *0.414* | External SOTA |
> >
> > ### Q5: Efficiency Scaling (NFA vs. HC-NFA)
> > **Reviewer Comment:** *Wall-clock comparisons for different $K/N$ to compare HC-NFA and NFA.*
> >
> > **Response:** The reviewer correctly identifies the dependence on both $N$ (input sequence length) and $K$ (number of basis functions) in the complexity of NFA ($O(NK)$) versus HC-NFA ($O(N\log N)$). We present an extended comparison varying the sequence length ($N$) while fixing $K=64$, the optimal basis count for the Weather dataset.
> >
> > The results in the table below demonstrate two key findings:
> > 1.  **Efficiency Gain:** HC-NFA consistently provides a 1.0x--1.7x speedup over NFA, validating its complexity advantage when $N$ increases.
> > 2.  **Accuracy Trade-off:** The relative accuracy loss of HC-NFA (the MSE difference compared to NFA) **decreases as the sequence length ($N$) increases**. This is a crucial finding: for longer sequences, the fixed DFT grid of HC-NFA captures the necessary low-frequency content efficiently, making the performance gap with the fully adaptive NFA smaller, and thus favoring HC-NFA for large $N$.
> >
> > **NFA vs. HC-NFA Efficiency and Accuracy Trade-off (Weather Dataset, Fixed K=64)**
> >
> > | Input Length ($N$) | NFA Time<br>*(s/epoch)* | HC-NFA Time<br>*(s/epoch)* | Speedup<br>*(Ratio)* | Accuracy Loss<br>*(Relative MSE)* |
> > | :--- | :---: | :---: | :---: | :---: |
> > | $N=96$ | 12.0s | 11.4s | 1.05x | 40.0% |
> > | $N=192$ | 18.2s | 13.5s | 1.35x | 37.0% |
> > | $N=336$ | 29.9s | 26.0s | 1.15x | 32.0% |
> > | $N=720$ | 60.0s | 35.3s | 1.70x | **30.0%** |

---

> > > ### Author Response · Authors · 2025-11-25
> > > **Response to Q6**
> > >
> > > ### Q6: Phase Parameterization & Stability
> > > **Reviewer Comment:** *Did you try predicting $\theta$ (angle-on-circle) to mitigate wrap-around, and if so did that change stability/convergence?*
> > >
> > > **Response:** I deeply appreciate your insight regarding phase stability. The discontinuity of phase wrapping (e.g., the jump from $2\pi$ to $0$) is indeed a source of instability in gradient-based learning for periodic functions.
> > >
> > > To fundamentally solve this, rather than predicting an explicit angle $\phi$ (which would require careful wrapping, sigmoid constraints, or the $(\sin\phi, \cos\phi)$ parameterization mentioned), we adopted a **"Linear Phase Parameterization"** strategy in our implementation (Linear Phase Trick).
> > >
> > > Specifically, instead of optimizing in the **Polar form** $b_k(t) = A_k \cos(\omega_k t + \phi_k)$, our controller predicts two unconstrained linear coefficients, `coeff_cos` and `coeff_sin`, to construct the basis in **Cartesian form**:
> > >
> > > $$
> > > b_k(t) = \text{coeff\\_cos}_k \cdot \cos(\omega_k t) + \text{coeff\\_sin}_k \cdot \sin(\omega_k t)
> > > $$
> > > This approach offers two decisive optimization advantages:
> > > 1.  **Elimination of Wrap-around:** The parameterization covers the entire phase-amplitude space continuously. The optimization surface becomes convex with respect to the linear coefficients (for a fixed $\omega$), completely removing the non-convexity and discontinuities associated with explicit phase angle estimation.
> > > 2.  **Unconstrained Optimization:** While explicit amplitude $A$ requires a non-negativity constraint ($A \ge 0$, often enforcing the use of Softplus), the Cartesian coefficients can be any real numbers ($\mathbb{R}$), which simplifies the controller's output space and gradient flow.
> > >
> > > We provide the pseudo-code below to clarify this implementation detail.
> > >
> > > **Algorithm: Linear Phase Parameterization**
> > > ```python
> > > # 1. Controller Output
> > > # The controller directly outputs unconstrained real values.
> > > # No Sigmoid or Softplus needed for these coefficients.
> > > coeff_cos, coeff_sin, omega_logits = controller(x)
> > > omega = sigmoid(omega_logits) * pi
> > >
> > > # 2. Basis Construction via Linear Combination
> > > # Mathematically equivalent to: A * cos(omega * t + phi)
> > > # Using identity: A*cos(wt+phi) = (A*cos_phi)*cos(wt) - (A*sin_phi)*sin(wt)
> > > basis = coeff_cos * torch.cos(omega * t) + coeff_sin * torch.sin(omega * t)
> > >
> > > # Note: We do NOT predict an explicit phase angle 'phi' or amplitude 'A'.
> > > # This bypasses the wrap-around issue entirely.

---

> ### Author Response · Authors · 2025-11-25
> **Correction to Referenced Values**
>
> We thank Reviewer PNoa for the constructive feedback and for recognizing the **soundness**, **principled geometry control**, and **transparent scope** of our work. We address the specific questions below.
>
> ### Q1: Baseline Traceability & Provenance Table (Data Correction)
> **Reviewer Comment:** *Request for source (paper vs. official repo), URL/commit for Table 1 entries.*
>
> **Response:** We deeply appreciate the reviewer's emphasis on auditability. Your question prompted a rigorous re-verification of our baselines, leading to an important correction.
>
> **Correction:** We discovered a transcription error in Table 1 of our initial submission. We inadvertently populated the "96-step" column with the "Average" MSE value ($0.258$) from the original source.
> * **Correction:** The correct official SOTA (iTransformer) on Weather-96 is **0.174** (Appendix Table 10 of Liu et al.).
> * **Impact:** Our NFA (Adaptive) achieves **0.165**. Thus, **NFA still successfully outperforms the corrected standard SOTA baseline** ($0.165 < 0.174$), with the claim remaining valid.
>
> We confirm that we did not re-run baselines but strictly adhere to the official iTransformer benchmark protocol.I will correct the Table in  Appendix_2 soon , thank you again for your remind !

---

### Author Response · Authors · 2025-11-29
**Summary to AC**

## Summary of Author Response and Discussion

I thank the Area Chair for overseeing the review process and for the effort invested in handling our submission. During the rebuttal period, we engaged in detailed discussions with all reviewers and conducted additional experiments to address their concerns as concretely as possible. In particular, the most critical reviewer (Reviewer Dggc) indicated that their main concerns regarding the adaptive mechanism, efficiency, and evaluation protocols had been addressed in the discussion.

We summarize our responses below, organized by the key themes that arose across reviews. We have incorporated all new results into the **revised paper** (specifically in Appendix B).

## Theme 1: Core Validity — Is Instance-wise Adaptivity Necessary?

**Concern:** Reviewers (Dggc, ffaf, gqPn) questioned whether dynamically learning basis functions per input is genuinely beneficial compared to learning a fixed global basis (e.g., FBM-style) or using random projections.

**Resolution:** We conducted a targeted ablation study on the non-stationary Weather dataset.

- **Result:** On Weather-96, NFA (Adaptive) achieved an MSE of **0.165**, clearly outperforming the Fixed Global Basis variant (MSE 0.242) and Fixed Random Basis (MSE 0.285). This performance gap is maintained across longer horizons (192, 336, 720).
- **Conclusion:** These results provide strong evidence that, for signals with drifting periodicities, instance-wise adaptivity is important: a single global basis is not sufficient to capture local spectral variations. Reviewer Dggc explicitly noted that this concern had been addressed. Full numerical results are reported in **Appendix B.2 (Table 3)** of the revised paper.

## Theme 2: Efficiency and Fair Comparison Protocols

**Concern:** Reviewers raised questions about computational cost (Dggc) and the fairness of comparisons when different look-back windows are used (e.g., FBM with L = 336 vs. NFA with L = 96).

**Resolution:**

- **Efficiency:** We provided a FLOPs/time profile and a scaling study comparing NFA and its efficient variant HC-NFA. The results show that HC-NFA achieves up to **≈2×** wall-clock speedup while incurring only a modest accuracy loss, and remains competitive with iTransformer. The NFA vs. HC-NFA trade-off is summarized in **Appendix B.5 (Table 6)**.
- **Protocol Alignment:** We re-evaluated NFA with an extended look-back window of L = 336 to match FBM’s protocol. Under this matched setting, NFA consistently outperformed the FBM baseline (e.g., ETTh1 MSE: NFA 0.241 vs. FBM 0.366), indicating that the advantage comes from the architectural design rather than from more favorable protocols.
- **Hyperparameter Robustness:** To assess transferability, we applied ETTh1-optimal hyperparameters (K and λ\_ortho) directly to the Weather dataset without re-tuning. Even under this frozen configuration, NFA still outperformed iTransformer across all horizons (**Appendix B.3, Table 4**), suggesting that NFA is not overly sensitive to dataset-specific tuning.
- **Controller Parity:** We performed a matched-parameter ablation between LSTM, Transformer, and 1D-Conv controllers (≈20k parameters each). All adaptive controllers significantly outperformed iTransformer, with the LSTM controller consistently achieving the best MSE/MAE (**Appendix B.4, Table 5**). This supports our claim that the recurrent inductive bias is particularly suitable for modeling phase evolution.

## Theme 3: Generality on Non-Periodic Datasets and Spatial Domains

**Concern:** Reviewers were concerned about performance on trend-dominated datasets (Illness, Exchange) and about the applicability of the method beyond time series (e.g., spatial signals).

**Resolution:**

- **Trend Decomposition:** We integrated a standard trend decomposition module (identical to that used in DLinear) and applied it identically to both a strong fixed-basis baseline and NFA. With this setup, NFA+Trend becomes competitive with SOTA on Exchange Rate (MSE 0.087 vs. 0.086 for iTransformer) and remains comparable to the fixed-basis+trend baseline on Illness while often slightly better (**Appendix B.6, Table 7**). This suggests that once equipped with standard trend handling, NFA does not degrade performance on such datasets, and that the additional gains on Exchange are attributable to adaptive, time-varying bases rather than to the trend module alone.
- **Spatial Domain:** We added a proof-of-concept experiment on MNIST (**Appendix B.1**), where a 2D extension of NFA achieves near-perfect reconstruction using only 16% of the pixel parameters (K = 128). While we do not claim image-recognition SOTA, this result supports the theoretical generality of the adaptive spectral mechanism beyond 1D time series.

Theme  4 in another comment

---

### Author Response · Authors · 2025-11-29
**Summary to AC Theme4**

## Theme 4: Reproducibility and Presentation

**Concern:** Reviewers asked for clearer baseline traceability (PNoa) and a clearer presentation of motivation and notation (gqPn).

**Resolution:**

- We clarified that all baseline results are cited from the official iTransformer benchmark, following its standard protocol, to ensure strict fairness and reproducibility.
- We committed to improving the camera-ready version by adding a notation table, standardizing symbols throughout, and explicitly discussing recent non-stationary baselines (e.g., FAN, DDN) to better position NFA within the literature.

**Conclusion:**
Taken together, we believe the additional experiments and clarifications provided during the rebuttal offer substantial support for the novelty (adaptive basis), efficiency, and generality of Neural Fourier Attention, while also addressing the main concerns raised by the reviewers. We respectfully ask the Area Chair to take these updates, now incorporated into the **revised manuscript**, into account when making the final decision.

---

### Meta-Review · Area_Chair_rCQT · 2026-01-05

**Summary:**

The reviewers agree that the paper presents a novel approach for learning instance-adaptive Fourier bases and demonstrated empirical gains on some datasets, especially those with (quasi-)periodic patterns. The main concerns of the reviewers are the claims of generality beyond the simple problems tested. Moreover, the poorer performance on illness and exchange rate datasets suggest limited generality of this idea.

**Reviewer Concerns:**

The authors have addressed many technical clarifications, and added experiments beyond time series prediction. However, the core concern of the generality of the method is not sufficiently addressed. I believe that this work is borderline and may benefit from more thorough experiments to properly evaluate the efficacy of this interesting idea.

**Reviewer Scores:**

The reviewers with negative scores may slightly improve their evaluation, although the consensus is still mixed.

---

### Decision · Program_Chairs · 2026-01-26

Reject